# CURRICULUM-AUGMENTED GFLOWNETS FOR mRNA SEQUENCE GENERATION

## ABSTRACT

Designing mRNA sequences is a major challenge in developing next-generation therapeutics, since it involves exploring a vast space of possible nucleotide combinations while optimizing sequence properties like stability, translation efficiency, and protein expression. While Generative Flow Networks are promising for this task, their training is hindered by sparse, long-horizon rewards and multi-objective trade-offs. We propose **Curriculum-Augmented GFlowNets (CAGFN)**, which integrate curriculum learning with multi-objective GFlowNets to generate *de novo* mRNA sequences. CAGFN integrates a length-based curriculum that progressively adapts the maximum sequence length guiding exploration from easier to harder subproblems. We also provide a new **mRNA design environment** for GFlowNets which, given a target protein sequence and a combination of biological objectives, allows for the training of models that generate plausible mRNA candidates. This provides a biologically motivated setting for applying and advancing GFlowNets in therapeutic sequence design. On different mRNA design tasks, CAGFN improves Pareto performance and biological plausibility, while maintaining diversity. Moreover, CAGFN reaches higher-quality solutions faster than a GFlowNet trained with random sequence sampling (no curriculum), and enables generalization to out-of-distribution sequences.

## 1 INTRODUCTION

Imagine a molecule that can be designed to instruct human cells to produce a protein of interest. Such is the promise of messenger RNA (mRNA), which has become a cornerstone of modern biotechnology (Pardi et al., 2018; Sahin et al., 2014). Designing *de novo* mRNA sequences, that encode a target protein and achieve optimality on particular properties of interest (Gustafsson et al., 2004; Kane, 1995; Mauger et al., 2019), is therefore of growing practical importance. This task can be framed as generating long, structured sequences under multiple, often competing objectives, which makes search and optimization challenging (Keeney & Raiffa, 1993; Zhang et al., 2023; Angermueller et al., 2020).

Because biological targets are diverse and downstream outcomes are difficult to predict, diversity is a central design criterion (Mullis et al., 2019). This need is amplified by the limited predictive power of inexpensive screening methods, such as in-silico simulations or *in vitro* assays. To maximize the likelihood that at least one candidate proves successful, it is therefore important that the proposed sequences broadly cover the different modes of the underlying "goodness" function that estimates future success. The search space in this scenario becomes large, and the best solutions often lie not at a single optimum but across a diverse Pareto front of trade-offs. Due to codon redundancy (see Appendix A), the number of synonymous nucleotide sequences that encode a protein grows exponentially with its length, and they differ dramatically in biological properties. For example, the SARS-CoV-2 spike protein ($\sim$1,273 amino acids) has on the order of $10^{632}$ possible synonymous sequences, most of which are suboptimal for practical use (Zhang et al., 2023). Therefore, the goal of mRNA sequence design is to generate diverse sequences that (i) preserve the desired protein sequence, (ii) satisfy biological constraints, and (iii) expose explicit trade-offs between competing objectives so practitioners can choose sequences suited to specific applications.

Generative methods that can efficiently explore this combinatorial landscape, propose diverse high-quality candidates, and expose trade-offs between objectives are therefore of significant scientific

and practical value. Generative Flow Networks (GFlowNets; Bengio et al., 2021) offer a principled framework for such tasks. Unlike standard reinforcement learning (RL) or maximum likelihood methods, GFlowNets learn policies that sample complete objects with probability proportional to a user-specified non-negative reward $R(x)$. This property naturally supports the discovery of diverse high-reward solutions rather than collapse in a single mode, making them attractive for sequence design tasks where diversity is as important as optimization (Jain et al., 2022b; Roy et al., 2023). Recent work has begun applying GFlowNets to molecules and biological sequences (Jain et al., 2022a; Zhu et al., 2023; Cretu et al., 2024; Koziarski et al., 2024), showing their potential for structured generation in scientific domains.

Despite these advantages, training GFlowNets on a real-world sequence design task like mRNA reveals two fundamental challenges. First, sequences are long: rewards are only observed at the terminal step, leading to extremely sparse credit assignment. While objectives such as Trajectory Balance (Malkin et al., 2022) reduce variance compared to the Flow Matching and Detailed Balance losses (Bengio et al., 2021), they remain fragile for very long horizons. Second, the design problem is inherently multi-objective: optimizing a fixed combination of objectives typically drives the search into a narrow region of the design space, obscuring alternative solutions that are valuable in practice.

Curriculum Learning (CL; Bengio et al., 2009) offers a natural strategy to mitigate these challenges by structuring training so the learner first masters simpler tasks before progressing to harder ones. Crucially, modern curricula do more than sort tasks by difficulty: they preferentially present tasks on which the model is making the fastest progress, i.e., where the slope of the learning curve is highest. For GFlowNets, this perspective addresses long-horizon difficulty by using an *adaptive, length-based curricula* that allocate tasks dynamically based on recent learning progress to focus training where it is most informative (Matiisen et al., 2019). Although CL is an established paradigm, its use to steer GFlowNets for sequential biological design is, to our knowledge, novel.

**Contributions.** In this paper, we introduce **Curriculum-Augmented GFlowNets (CAGFN)**, a principled integration of Curriculum Learning with multi-objective GFlowNets for mRNA sequences. Although motivated by mRNA design, the method is applicable to a wide range of sequential generative tasks with long sequences and multi-objective trade-offs.

In the next sections:

- We introduce a *new* mRNA design environment, compatible with the *torchgfn* library (Lahlou et al., 2023), for GFlowNets that, given a target protein sequence, generates diverse synonymous mRNA candidates that optimize multiple biologically motivated objectives. We illustrate the generation process of mRNA sequences in Figure 1.

- We propose adaptive, length-based curricula that preferentially present tasks with high learning progress to accelerate training and stabilize credit assignment.

- We show that Curriculum-Augmented GFlowNets (CAGFN) improve training and make the model useful for generating diverse sequences for different proteins, demonstrating a scalable and principled path toward generative modeling in complex biological and scientific domains.

**Novelty.** To our knowledge, this is the first systematic study that combines Curriculum Learning, in particular adaptive curricula that prioritize tasks with the steepest learning-curve slope, with multi-objective GFlowNets for mRNA sequence design. Our contributions are both methodological (curriculum mechanisms tailored to GFlowNets) and empirical (a biologically motivated environment for mRNA sequence design).

## 2 RELATED WORK

Generative Flow Networks (GFlowNets) have been applied to a range of scientific design problems, including molecules and biological sequences (Bengio et al., 2021; 2023; Jain et al., 2022a; 2023; Cretu et al., 2024; Koziarski et al., 2024). Recent extensions address multi-objective optimization and domain constraints, highlighting their potential for structured generation tasks.

To our knowledge, GFlowNets have not yet been applied to mRNA sequence design, which poses distinct challenges due to codon redundancy, biological constraints, and long sequences with

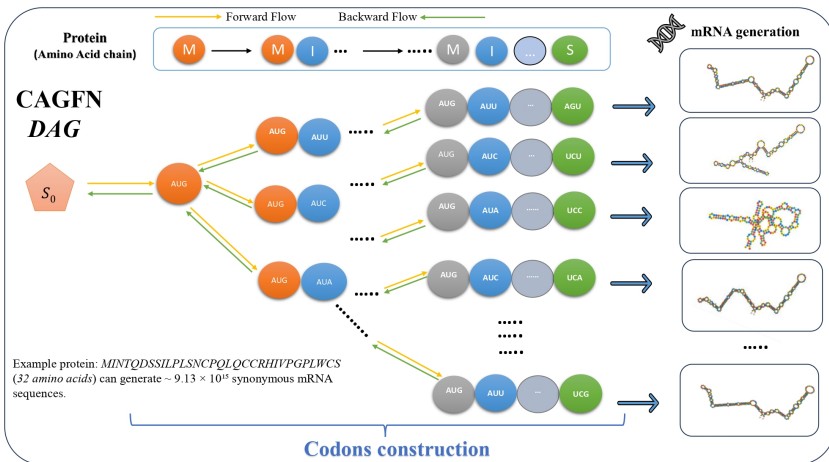

Figure 1: **Directed acyclic graph (DAG)** representation of the mRNA sequence design process. Each node corresponds to a codon selection step, and edges represent possible transitions, illustrating the sequential construction of mRNA sequences. This DAG (tree in this case) forms the basis for the GFlowNet to explore diverse and high-quality sequences.

sparse rewards. These factors motivate our integration of curriculum learning with multi-objective GFlowNets. A more detailed review of related work, including multi-objective GFlowNets and curriculum learning, is provided in Appendix I.

## 3 BACKGROUND

### 3.1 MRNA SEQUENCE DESIGN

Messenger RNA (mRNA) is a linear polymer built from nucleotide alphabet $\Sigma = \{A, U, G, C\}$. Proteins are sequences of amino acids and each amino acid is encoded by a *codon*, a contiguous triplet of nucleotides. The standard genetic code maps the set of 64 codons onto 20 amino acids plus stop signals. Because several codons can encode the same amino acid (synonymous codons), the mapping is degenerate (see definition and examples in Appendices A). This redundancy creates a large combinatorial set of possible mRNA sequences that all encode the same protein, and motivates algorithmic search and generative approaches for this design task.

**Design objectives.** To maximize the chances that at least one of the candidates will work at the end, it is important for these candidates to cover as much as possible the modes of a *goodness* function that estimates future success. Diversity-aware methods trade breadth for targeted exploration of multiple promising modes, improving the probability of downstream success and accelerating the overall design cycle (Pyzer-Knapp, 2018; Terayama et al., 2021). Generative models that can exploit the combinatorial structure in these spaces have the potential to speed up the design process for such sequences (see Appendix C for further motivation). mRNA design is inherently multi-objective: beyond preserving the protein sequence, we aim to optimize biological properties such as translation efficiency, stability and protein expression, (Sahin et al., 2014). Three commonly used objectives for mRNA design (Sahin et al., 2014; Pardi et al., 2018) are: **Codon Adaptation Index (CAI)**: A measure of how well the sequence conforms to species-specific codon usage preferences. Higher CAI typically leads to more efficient translation (Sharp & Li, 1987; Lee, 2018). **Minimum Free Energy (MFE)**: The (negative) folding free energy of the mRNA's secondary structure (Zuker & Stiegler, 1981). A lower MFE (more negative) indicates a more stable folded structure. **GC content**: The fraction of G or C nucleotides in the sequence. High GC-content generally correlates with increased mRNA stability and translation efficiency (Courel et al., 2019), in particular, optimizing GC-content has a similar effect to optimizing codon usage (Zhang et al., 2023).

## 3.2 GFLOWNETS

GFlowNets (Bengio et al., 2021; 2023) are a class of probabilistic generative models that learn a stochastic policy to construct compositional objects $x \in \mathcal{X}$ (e.g., graphs, sequences) by sampling trajectories $\tau$ in a weighted directed acyclic graph (DAG) $G = (\mathcal{S}, \mathcal{E})$. Construction begins at an initial empty state $s_0$ and proceeds by iteratively sampling actions $a \in \mathcal{A}$ according to a forward policy $P_F(\cdot \mid s)$ until a terminal state $x = s_n$ is reached. The aim is to learn $P_F$ so that the marginal probability $\pi(x)$ of sampling $x$ under the forward policy is proportional to a given non-negative reward $R(x)$: $\pi(x) \propto R(x)$, $Z = \sum_{x \in \mathcal{X}} R(x)$, where $Z$ is the partition function.

For a trajectory $\tau = (s_0, a_0, s_1, \ldots, a_{n-1}, s_n = x)$, the GFlowNet forward and backward policies are defined by: $P_F(\tau) = \prod_{t=0}^{n-1} P_F(a_t \mid s_t)$, $P_B(\tau) = \prod_{t=1}^{n} P_B(a_{t-1} \mid s_t)$. Let $\pi(x)$ be the marginal likelihood of sampling trajectories terminating in $x$ by sampling actions from the forward policy is $\pi(x) = \sum_{\tau \to x} P_F(\tau)$, where the sum is over all trajectories that terminate at $x$. The GFlowNet learning problem involves learning a policy $P_{F,\theta}$ such that the induced marginal $\pi_\theta(x) = \sum_{\tau \to x} P_{F,\theta}(\tau)$ approximates the target $\frac{R(x)}{Z}$, where $Z$ is the unknown partition function, for all $x \in \mathcal{X}$.

In this work, we adopt the *sub-trajectory balance* (Sub-TB) objective of Madan et al. (2023). A more formal treatment and objectives (TB/SubTB) are provided in Appendix D. We refer the reader to Malkin et al. (2022); Madan et al. (2023); Bengio et al. (2021; 2023) for a more thorough introduction to GFlowNets.

# 4 METHODOLOGY

## 4.1 GFLOWNET-BASED GENERATION OF MRNA SEQUENCES

We consider the problem of searching over a space of mRNA sequences that encode a target protein $a = (a_1, \ldots, a_L)$ and find codon sequences that maximize a multi-objective reward function $R(x)$ capturing design objectives (§ 3.1). A codon sequence $x = (c_1, \ldots, c_L)$ corresponds to a set of codons where each codon $c_i \in \mathcal{C}(a_i)$ encodes an amino acid $a_i$. Here, $\mathcal{C}(a_i)$ denotes the set of synonymous codons encoding the same amino acid $a_i$. Thus, the feasible design space is $\mathcal{X} = \prod_{i=1}^{L} \mathcal{C}(a_i)$. Since most of amino acids have multiple synonymous codons, $|\mathcal{X}|$ grows exponentially with $L$.

To investigate the capacity of GFlowNets for mRNA design, we introduce a new environment **CodonDesignEnv**, compatible with the torchgfn library (Lahlou et al., 2023), (details are in Appendix J), which models the task of generating mRNA sequences for a specific protein.

**Environment, States, and Actions.** The environment models mRNA sequence construction as a trajectory through states $s_t = (c_1, \ldots, c_t)$, $s_0$ is the empty sequence and $s_t$ a partially constructed mRNA sequence. A terminal state $x = s_L$ corresponds to a complete mRNA sequence encoding the target protein. The action space consists of the 64 codons plus a special exit action, yielding $|\mathcal{A}| = 65$. Transitions are deterministic and constrained by the genetic code. At step $t < L$, the forward action set is dynamically masked to only allow synonymous codons for the $(t+1)$-th amino acid. At $t = L$, only the exit action is valid, forcing termination. These dynamic masks ensure that every trajectory from $s_0$ to the terminal sink state $s_f$ corresponds to a valid mRNA sequence, and preserve the encoding of the original protein sequence. The resulting state-action graph is a directed acyclic graph (DAG) 1 aligned with the codon redundancy of the genetic code, providing a natural fit for flow-based generative modeling with GFlowNets. Rewards are assigned only to terminal states and capture biologically motivated objectives (§3.1). For a complete mRNA sequence $x$, the reward is $R(x) = w^\top \phi(x)$, where $\phi(x)$ is a vector of normalized objectives and $w \in \mathbb{R}^3$ are objective weights. By adjusting $w$, one can bias the GFlowNet toward different Pareto-optimal trade-offs, promoting diversity and mode coverage across the design space. Our framework supports both (i) unconditional training, where the weights of the reward are fixed and define a single trade-off between objectives, and (ii) conditional training, where the desired objective weights are provided at training and inference, enabling controlled exploration of the Pareto front.

**Unconditional Generation.** In the unconditional setting, we fix the objective weight vector $w$ during training so the GFlowNet learns a fixed reward function $R(x \mid w) = w^\top \phi(x)$. The model is trained (see Algorithm 1 in Appendix E) to assign flows over complete trajectories such that the sampling probability of the trajectory is proportional to its reward $\pi(x) \propto R_w(x)$, encouraging the sampler to visit high-reward regions while preserving mode coverage. At inference, we sample complete trajectories from the learned forward policy, producing a diverse pool of candidate mRNA sequences whose sampling probability reflects their composite reward under the chosen weights. This contrasts with optimization-only methods (Williams, 1992; Schulman et al., 2017; Holland, 1992; Snoek et al., 2012) that collapse to a single optimum.

**Conditional Generation.** As the biological objectives are often imperfect proxies for mRNA properties of interest, we aim to generate diverse Pareto-optimal solutions for each sub-problem $\max_{x \in \mathcal{X}} R(x \mid w)$. Also, fixing a single set of weights for the GFlowNet at a time prevents the model from exploiting the shared structure present between these related sub-problems. Bengio et al. (2021) extend standard GFlowNets to learn a single conditional policy that simultaneously models a family of distributions indexed by a conditioning variable. Each weight vector $w \in \mathcal{W}$ induces a different reward function $R(x \mid w)$ over terminal states $x \in \mathcal{X}$. The conditioning information $w$ is then available at every state $s \in \mathcal{S}_w$.

Denote by $P_F(\cdot \mid s, w)$ and $P_B(\cdot \mid s', w)$ the conditional forward and backward policies respectively, and let the conditional partition function be $Z(w) = \sum_{x \in \mathcal{X}} R(x \mid w)$. The goal of a reward-conditional GFlowNet is to learn $P_F(\cdot \mid s, w)$ (parameterized, e.g., by a neural network $P_{F,\theta}(a \mid s, w)$) such that the marginal probability of sampling a terminal state $x$ under the forward policy satisfies $\pi(x \mid w) \propto R(x \mid w)$. Exploiting the shared structure across $w$-conditioned reward functions allows a single conditional policy to model the entire family of sub-problems and, importantly, to generalize to different weight vectors $w$.

### 4.2 Curriculum Learning

Beyond weights generalization, we want the model to generalize across different type of proteins (of different lengths). To do so, we introduce a Curriculum-Augmented GFlowNets (CAGFN), an integration of Curriculum Learning with a multi-objective GFlowNet.

The core idea is to expose the GFlowNet to multiple training tasks, so that the model incrementally acquires codon-level and local-structure patterns on different type of proteins. Practically, we partition training into length intervals and advance the curriculum according to a strategy that systematically adapts learning tasks of different difficulty based on GFlowNets learning. It allows the same parameterized policy to refine and extend learned representations as task complexity grows. This length-aware curriculum encourages hierarchical feature learning, improves sample efficiency, and reduces optimization instability when exploring extremely large combinatorial spaces. We use the following task set of amino-acid (AA) length intervals in our experiments:

$$\text{Tasks} = \{[l_1, l_2], [l_3, l_4], [l_5, l_6], [l_7, l_8], [l_9, l_{10}]\} \text{ (AA)}, \tag{1}$$

Each pair $(l_i, l_{i+1})$ defines an interval of protein lengths from which we sample during training; given a sampled length, the corresponding protein sequence is then drawn from our pool of available proteins. During each curriculum stage, the sampling is restricted to the proteins within the current interval.

Curriculum Learning (CL) describes training procedures that expose a learner to a sequence of tasks of increasing difficulty (Bengio et al., 2009). We use an automatic, adaptive curriculum where a "Teacher" selects tasks based on the "Student's" learning progress, rather than a hand-designed schedule (Matiisen et al., 2019; Willems et al., 2020).

#### 4.2.1 Teacher-Student CL: a formal recipe

We follow the teacher-student (TSCL) paradigm (Matiisen et al., 2019) in which the Teacher maintains a per-task performance signal and uses *learning progress* to bias sampling of tasks. For each task index $i \in \{1, \ldots, K\}$ and discrete training step $t$ define a per-task metric $m_{i,t}$. The Teacher forms a smoothed learning-progress estimate $\text{LP}_{i,t}$ from the discrete changes $\Delta m_{i,t} =$

$m_{i,t} - m_{i,t-1}$. A common and stable choice for the learning-progress estimate $\text{LP}_{i,t}$ is an exponential moving average (EMA):

$$\mathcal{LP}_{i,t} = (1 - \beta)\,\mathcal{LP}_{i,t-1} + \beta\,(m_{i,t} - m_{i,t-1}), \tag{2}$$

with smoothing hyperparameter $\beta \in (0, 1]$ (e.g. $\beta \approx 0.01\text{--}0.1$). The Teacher then converts LP to a sampling distribution over tasks. A robust, positivity-enforcing rule is:

$$P(i \mid t) = \frac{\max\!\big(0, \mathcal{LP}_{i,t}\big) + \varepsilon}{\sum_{j=1}^{K}\big(\max\!\big(0, \mathcal{LP}_{j,t}\big) + \varepsilon\big)}, \tag{3}$$

where $\varepsilon > 0$ is a small floor to ensure exploration. We further describe and compare different curriculum configurations in Appendix F.1.

**Choice of per-task metric.** The metric $m_{i,t}$ should reflect the Student's progress on task $i$. The design is robust to the specific metric as long as it (i) changes as the Student learns, and (ii) is reasonably smooth or smoothed by EMA to reduce noise. We choose for this work the average reward over generated sequences, as a per-task metric, since it reflects the performance of our GFlowNets (the Student model).

$\mathcal{LP}$-**based selection.** Selecting tasks with highest positive $\mathcal{LP}$ focuses training on regions where the Student still makes rapid gains, accelerating sample efficiency and directing compute to the frontier of competence. Negative $\mathcal{LP}$ (decline in performance) can be used to trigger replay and reduce forgetting (Matiisen et al., 2019). This approach necessitates a dynamic training setup where the environment's constraints change with each task. We initialize a single, shared GFlowNet model. The central principle of our methodology is parameter sharing. The model does not learn a separate policy for each protein sequence, instead, it learns a single, conditional policy that can generalize across the entire task distribution. This forces the model to learn the underlying principles of codon selection that lead to high rewards, rather than memorizing solutions for specific proteins.

The complete procedure is detailed in Algorithm 2 in Appendix G. In summary, it proceeds as follows:

- **Initialization:** We define the set of curriculum tasks $T$ (cf. Eq.1), each associated with a pool of protein sequences $S_i$ [1]. The GFlowNet policy $\pi_\theta$, is initialized, along with the task-sampling distribution $P$, which initially assigns equal probability to all tasks.

- **Task Sampling:** At each training step (Alg. 2, lines 3–5), the Teacher samples a task index $k \sim P$. A protein sequence $p_{\text{seq}}$ is then drawn from the corresponding pool $S_k$, and a task-specific environment $\mathcal{E}$ is instantiated for that sequence.

- **GFlowNet Training:** The shared policy $\pi_\theta$ is trained for $I_{\text{task}}$ iterations on this environment (Alg. 2,lines 6–11). At each iteration, a weight vector $\boldsymbol{w}$ is sampled, trajectories are generated, and the parameters $\theta$ are updated via the SubTB loss. Afterward, the environment is discarded, and a new task is sampled.

- **Evaluation and Adaptation:** At fixed intervals $I_{\text{eval}}$ (Alg. 2, lines 12–28), the Student is evaluated on all tasks using a fixed evaluation weight vector $\boldsymbol{w}_{\text{eval}}$. For each task $t_j$, we compute the mean reward across generated sequences and compare it to the previous evaluation. The difference defines a learning progress signal $\Delta m_j$, which updates $\mathcal{LP}_j$. Tasks with higher $\mathcal{LP}_j$ are assigned greater sampling probability in the following taring step, updating $P$ to favor tasks where the Student shows improvement.

## 5 EXPERIMENTS

All hyperparameter choices for curriculum, policy architecture, and optimization are detailed in Appendix F (Tables 3–5). We begin our experiments by analyzing the model in an unconditional setting to establish a baseline, then proceed to the primary conditional generation task with and without CL.

---

[1]We created the pool of protein sequences sets for each tasks from the CodonTransformer dataset (Fallah-pour et al., 2025)

## 5.1 Preliminary Study

Although our primary objective is conditional mRNA generation, we first conducted exploratory experiments in an unconditional setting to validate the core GFlowNet architecture. We used a MOGFN (Jain et al., 2023) with fixed weights to generate mRNA sequences for a small protein sequence task. The model produced diverse sequences with competitive scores across key biological objectives, and to assess the validity of our generated sequences, we compared them with the corresponding natural available mRNA sequence to check the range of our metrics, for example, the best generated sequence (vs. the natural one from our dataset in Appendix F): achieved a GC content of $53.78\%$ (vs. $45.33\%$), MFE of $-79.23$ (vs. $-68.20$), CAI of $0.63$ (vs. $0.54$), therefore outperforming the natural sequence in all metrics. The Pareto front of the generated sequences is shown in Figure 2a, while Figure 2b illustrates the distributions of rewards of the generated sequences.

However, real-world mRNA design requires adapting to different weights configurations, focusing only on unconditional generation fails to capture the full Pareto front of biologically relevant solutions. In the remainder of this paper, we therefore focus exclusively on the conditional GFlowNet.

We next assess the benefits of conditioning over the objectives weights. To do so, we compared on a small protein sequence task, a multi-objective GFlowNet conditioned on the objectives weights (the weights are sampled during training from a Dirichlet distribution $Dirichlet(1, 1, 1)$), and an unconditional GFlowNet, Figure 3 illustrates the results, with a fixed objectives weigths (the reward is weighted sum of objectives given these weights).

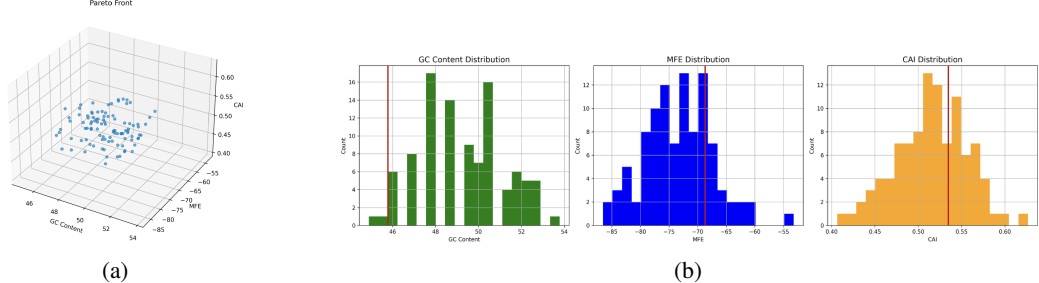

(a)  (b)

Figure 2: Unconditional mRNA sequence generation with GFlowNets under fixed objective weights $[0.3, 0.3, 0.4]$. **(a)** Pareto front. **(b)** Distribution of reward metrics: The spread demonstrates that the model explores diverse regions of the design space rather than collapsing to a single mode. Vertical lines represent the natural sequence scores.

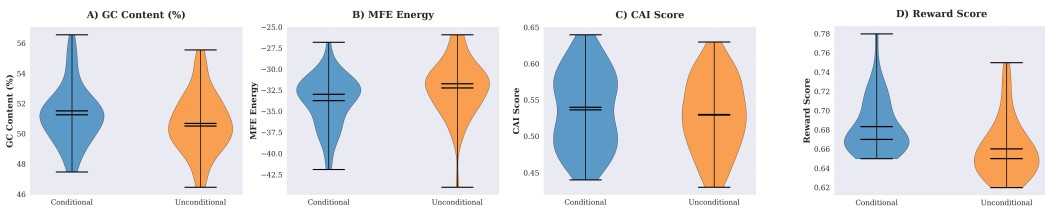

Figure 3: **Conditional Vs Unconditional mRNA generation results.** Metrics distribution across mRNA sequences of a small protein of interest ($\sim$35AA). More details in Figure 6 of Appendix H.

## 5.2 Comparison with Reinforcement Learning

To evaluate its effectiveness, we compare our model against RL baselines for mRNA design. We consider the closely related MOReinforce (Lin et al., 2022) as a baseline, as in the MOGFN work (Jain et al., 2023). We did not compare our method with MARS (Xie et al., 2021) or standard MCMC approaches for mRNA design. MARS is specifically designed for combinatorial optimization in small-molecule design, relying on iterative local mutations and chemical property predictors, which do not directly translate to the multi-objective nature of mRNA design. Similarly, traditional MCMC methods, while theoretically applicable, suffer from poor scalability in extremely large com-

binatorial spaces such as mRNA sequences and tend to produce low-diversity solutions concentrated in local modes (Terayama et al., 2021; Pyzer-Knapp, 2018). Instead, we compared our method to PPO (Schulman et al., 2017), a RL approach that has been successfully applied to sequence generation tasks.

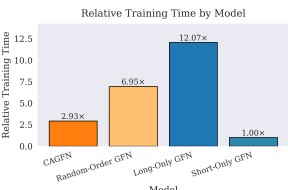

Figure 4: Training differences.

| Method | Reward (↑) | Uniqueness (↑) | Diversity (↑) |
|---|---|---|---|
| MOReinforce | **0.7** | 1 | 19 |
| PPO | 0.51 (±0.02) | 12 | 30 |
| MOGFN (w/o CL) | 0.59 (±0.064) | **100** | **31** |
| CAGFN | 0.59 (±0.066) | **100** | **31** |

Figure 5: Averaged metrics across 100 generated sequences for a small protein task.

As shown in Table 5, RL baselines like MOReinforce (Lin et al., 2022) and PPO (Schulman et al., 2017) suffered from mode collapse and low diversity, generating repetitive sequences (88% duplicates for PPO). In contrast, and consistent with prior work (Jain et al., 2023), our GFlowNet approach maintained high rewards while generating fully unique and diverse candidates, highlighting its suitability for this exploration task.

## 5.3 CURRICULUM LEARNING BENEFITS

To isolate the benefits of the CL strategy, we compare four distinct training strategies using the same conditional multi-objective GFlowNets (MOGFN) (Jain et al., 2023) architecture:

- **Short-Only GFN (SGFN)**: A baseline model trained exclusively on a dataset of short protein sequences (30–60 amino acids). This tests the performance of a specialized model within its domain.

- **Long-Only GFN (LGFN)**: A second baseline trained exclusively on long protein sequences (125–180 amino acids). This model is used to assess the difficulty of learning complex, long-range dependencies from scratch.

- **Random-Order GFN (ROFN)**: A control model trained using the full dataset of proteins spanning all length ranges. However, sequences are presented in a random, unstructured order. This baseline helps determine if performance gains come from data diversity alone, rather than the structured curriculum.

- **Curriculum-Augmented GFN (CAGFN) (Ours)**: The proposed model, trained on the full dataset using a curriculum that organizes tasks by increasing sequence length. This strategy is designed to enable the model to learn hierarchical patterns and generalize effectively.

**Loss Dynamics and Curriculum Progression.** To better understand optimization behavior, we tracked both (i) the training loss and (ii) the distribution of protein sequence lengths sampled during training across the four regimes described in § 5.3, and we reported, in the Appendix in Figure 11, 10 curriculum dynamics. Moreover, as shown in Figure 4, CAGFN substantially reduces training time compared to both ROFN and LGFN, while maintaining high-quality sequence generation. Concretely, CAGFN trains $4\times$ faster than LGFN and $2.4\times$ faster than ROFN. This demonstrates that CL not only improves sample efficiency but also eases optimization when dealing with longer and more complex protein sequences. See Appendix H.1 for the full set of loss curves.

To evaluate the effectiveness of our CL strategy, we test the trained models on unseen protein sequences from both the small (30-60 AA) and medium (85-120 AA) length ranges. We compare the CAGFN against the three baselines.

As shown in Table 1, the results demonstrate that the CL approach yields a model that generates high-quality mRNA sequences and achieve a high Pareto performance. On the set of small proteins, the CAGFN achieves performance that is competitive with, and often superior to, the Specialist Short-Only GFN. This finding shows that training on a broad curriculum does not compromise performance on the simpler tasks learned early on. On medium-length proteins, CAGFN consistently matches or exceeds the performance of the other models. The comparison against the ROGFN is particularly revealing: while both models were trained on the same diverse dataset, and generates both

high-quality sequences, the superior Pareto efficiency of CAGFN model indicates that the structured order of task presentation is a key ingredient. It allows the model to build hierarchical representations of codon selection patterns that generalize better than those learned from a random sequence of tasks.

Table 1: TopK (K = 50) Reward and Diversity across generated sequences for different types of protein sequences. We use the Top-K Diversity and Top-K Reward metrics (Bengio et al., 2021). Standard deviations of sample scores are shwon between parentheses.

| Model | TopK Reward (↑) | | | | | TopK Diversity (↑) | | | | |
|---|---|---|---|---|---|---|---|---|---|---|
| | 0 | 1 | 2 | 3 | 4 | 0 | 1 | 2 | 3 | 4 |
| *(25–60 AA)* | | | | | | | | | | |
| SGFN | 0.58 (±0.01) | 0.52 (±0.05) | 0.55 (±0.02) | 0.64 (±0.02) | 0.67 (±0.02) | 24 (±3.2) | 32 (±3.1) | 37 (±3.0) | 40 (±3.5) | 36 (±3.3) |
| CAGFN | 0.61 (±0.06) | **0.58** (±0.05) | **0.62** (±0.05) | 0.69 (±0.06) | **0.73** (±0.01) | 25 (±3.3) | 32 (±3.2) | 38 (±3.8) | **41** (±3.6) | 37 (±4.0) |
| ROGFN | 0.58 (±0.02) | 0.53 (±0.02) | 0.55 (±0.02) | 0.67 (±0.02) | 0.70 (±0.02) | **25** (±3.3) | **32** (±3.4) | **38** (±3.2) | 40 (±3.5) | 37 (±3.3) |
| LGFN | **0.62** (±0.03) | 0.57 (±0.03) | **0.61** (±0.02) | **0.70** (±0.03) | **0.73** (±0.03) | 24 (±3.1) | 30 (±3.2) | 36 (±3.3) | 38 (±3.5) | 35 (±3.3) |
| *(85–120 AA)* | | | | | | | | | | |
| SGFN | 0.55 (±0.03) | 0.64 (±0.02) | 0.69 (±0.02) | 0.71 (±0.02) | 0.65 (±0.03) | 37 (±3.0) | **40** (±3.2) | **36** (±3.7) | 34 (±3.4) | 39 (±3.1) |
| CAGFN | 0.55 (±0.06) | 0.65 (±0.06) | 0.70 (±0.05) | 0.71 (±0.08) | **0.72** (±0.01) | 37 (±3.2) | **40** (±3.3) | **36** (±3.9) | 34 (±3.5) | **40** (±3.2) |
| ROGFN | 0.56 (±0.02) | 0.67 (±0.02) | 0.70 (±0.02) | **0.72** (±0.02) | 0.66 (±0.02) | 38 (±3.4) | **40** (±3.3) | **36** (±3.2) | **36** (±3.7) | 39 (±3.0) |
| LOGFN | **0.60** (±0.03) | **0.70** (±0.03) | **0.73** (±0.02) | 0.69 (±0.03) | **0.72** (±0.03) | 37 (±3.1) | 38 (±3.2) | 34 (±3.5) | 35 (±3.4) | 38 (±3.3) |

Indexes 0–4 denote protein task indices.

Table 2: Pareto Performance over 50 generated sequences across models.

| Model | Pareto Performance (↑) | | | | |
|---|---|---|---|---|---|
| | 0 | 1 | 2 | 3 | 4 |
| *(25–60 AA)* | | | | | |
| SGFN | 0.08 | 0.06 | 0.09 | 0.16 | 0.19 |
| Random-Order GFN | **0.13** | **0.15** | 0.17 | 0.17 | 0.21 |
| CAGFN | 0.07 | 0.12 | **0.21** | **0.21** | **0.22** |
| Long-Only GFN | 0.05 | 0.13 | 0.21 | 0.21 | 0.20 |
| *(85–120 AA)* | | | | | |
| SGFN | 0.05 | 0.21 | 0.13 | 0.15 | 0.16 |
| Random-Order GFN | 0.09 | 0.11 | **0.28** | 0.19 | **0.22** |
| CAGFN | **0.15** | **0.21** | 0.19 | **0.21** | 0.19 |
| LGFN | 0.05 | 0.13 | 0.13 | 0.21 | 0.20 |

Indexes 0–4 denote protein task indices.

The CAGFN, in table 2 shows a good coverage of the Pareto front (up to 22 sequences) across all different types of proteins.

**OAZ2 protein.** We also studied the performance on one particular protein, ornithine decarboxylase antizyme 2 (OAZ2). OAZ2 is a protein that helps regulate the levels of certain small molecules called *polyamines* inside human cells. It ensures that the cell keeps the right balance of them and doesn't make too much. We present in Figure 8 or Appendix H the distribution of reward objectives across the four models. The results, shown in Figure 8 reveal that our CAGFN consistently matches or exceeds baselines, and generates high-quality mRNA sequences.

Results for a medium-length protein task, shown in Figure 7 of Appendix H, further confirm that CAGFN shifts distributions toward more preferable values.

# 6 CONCLUSION

Motivated by the growing need for efficient and reliable mRNA therapeutics and vaccines, we addressed the problem of mRNA sequence design. The task focuses on optimizing codon sequences that encode the same protein but differ in biological properties. We introduced Curriculum-Augmented Generative Flow Networks (CAGFN), a multi-objective GFlowNet enhanced with a teacher-driven curriculum. Building on a MOGFN formulation, we designed a TSCL-style teacher that schedules tasks according to learning progress, enabling the model to generate diverse and high-quality mRNA sequences across proteins of varying lengths. Our experiments showed that CAGFN consistently improves upon non-curriculum GFlowNet baselines by achieving higher rewards, better objective trade-offs, faster convergence, and broader Pareto-front coverage. A limitation of the proposed method is that the optimization relies on computational proxies, which may not directly correlate with in vivo translation efficiency and protein expression. **Future work** may extend the curriculum framework to adapt along multiple axes of difficulty (e.g, each round of the curriculum could focus on optimizing a single objective).

## REPRODUCIBILITY STATEMENT

To ensure complete reproducibility of our results, we provide implementation details and make all resources publicly available in `https://anonymous.4open.science/r/GFN_for_mRNA_design-FCC3/`. The complete source code includes the different training modes and the dataset we used for protein/mRNA task generation. The algorithms used are outlined in the appendices. The final set of hyperparameters used to reproduce the reported results are discussed in Appendix F, and provided as default values in our code. Additionally, we provide evaluation scripts for the comparison validation and comprehensive testing procedures that allow researchers to reproduce all experimental results reported in Section 5.

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

## A  mRNA Design Problem

**Codon Redundancy.**    The genetic code is redundant: multiple codons can encode the same amino acid. For example, the amino acid **Leucine (Leu)** is represented by six codons (UUA, UUG, CUU, CUC, CUA, CUG), while **glycine** has four (GGU, GGC, GGA, GGG). This allows different mRNA sequences to yield the same protein. As an illustration, a short peptide Met–Leu–Gly may be encoded by the mRNA sequence `AUG--UUA--GGU` or alternatively by `AUG--CUG--GGC`, both producing the identical protein despite differing at the nucleotide level. Exploiting codon redundancy is central to mRNA sequence design, as it provides a vast combinatorial space in which sequences can be optimized for multiple objectives while preserving the encoded protein.

## B  mRNA Design Problem

To clarify the nature of the mRNA design problem, we provide a simple example that highlights the role of codon redundancy and the search space of feasible mRNA sequences.

**Example : A short protein sequence.**    Consider designing an mRNA sequence for a short protein consisting of three amino acids: Methionine–Phenylalanine–Lysine **(Met–Phe–Lys)**.

- Met (M) can only be encoded by the codon `AUG`.
- Phe (F) can be encoded by `UUU` or `UUC`.
- Lys (K) can be encoded by `AAA` or `AAG`.

The feasible design space for this protein is therefore:

$$\{\texttt{AUG UUU AAA, AUG UUU AAG, AUG UUC AAA, AUG UUC AAG}\}.$$

All four codons sequences produce the same protein, but differ in codon usage patterns. Choosing among them involves evaluating biological properties such as GC content, minimum free energy (MFE), and codon adaptation index (CAI). This simple case illustrates the exponential growth of the design space as protein length increases, underscoring the need for generative models like GFlowNets to explore it effectively.

## C  Motivation for diversity

Diversity is essential in biological sequence design because targets, mechanisms of action, and structural contexts are highly heterogeneous and often change over time (Mullis et al., 2019). Cheap preclinical screens (in silico or in vitro) are imperfect proxies for eventual performance in wet-lab experiments, so concentrating effort on a single 'best' sequence risks failure if that sequence aligns with biases or blind spots of the surrogate assays. By contrast, proposing a diverse set of high-quality candidates increases the chance that at least one design occupies a different mode of the true, and unknown, success landscape, providing robustness to model misspecification and experimental noise. Practically, this is critical when searching astronomical, combinatorial spaces (e.g., on the order of $10^{60}$ possible sequences) under tight experimental budgets: diversity-aware methods trade breadth for targeted exploration of multiple promising modes, improving the probability of downstream success and accelerating the overall design cycle (Pyzer-Knapp, 2018; Terayama et al., 2021). Generative models that can exploit the combinatorial structure in these spaces have the potential to speed up the design process for such sequences.

## D  Background on GFlowNets

GFlowNets (GFNs) are a family of probabilistic models that learn a stochastic policy to generate compositional objects, such as a graph describing an mRNA sequence, through a sequence of steps, with probability proportional to their reward $R(x) \geq 0$ (Bengio et al., 2021).

We consider a directed acyclic graph $(\mathcal{S}, \mathcal{A})$, where nodes $\mathcal{S}$ represent states and edges $\mathcal{A}$ represent actions. If $(s, s') \in \mathcal{A}$, we say that $s$ is a parent of $s'$ and that $s'$ is a child of $s$. There is a unique state $s_0$ with no parents that we name *source state* and a unique state $s_f$ with no children that we name *sink*

*state*. We refer to the parents of the *sink state* as terminating states $\mathcal{X} = \{s \in \mathcal{S} \mid (s, s_f) \in \mathcal{A}\}$. A complete trajectory $\tau = (s_0, s_1, ..., s_n, s_{n+1} = s_f)$ is a sequence of states that start with the *source state* and ends with the *sink state*.

We also consider a reward function $R : \mathcal{X} \to \mathbf{R}$ such as $\forall x \in \mathcal{X}, R(x) > 0$. By convention, we can also consider that the reward for non terminating states is zero $\forall s \in \mathcal{S} \setminus \mathcal{X}, R(s) = 0$. The goal of the GFlowNet is to learn to sample terminating states proportionally to the reward function $P_T(x) \propto R(x)$. To achieve this, GFlowNets start from the initial state (for example, empty sequence or molecule in biological setting, initial coordinates in hyper-grid environment...) and iteratively sample actions to move to the following states.

Formally, we consider a non-negative flow function $F : \mathcal{A} \to \mathbf{R}$. The goal of the GFlowNet framework is to learn such a flow function that satisfies the following flow matching (FM) and reward matching constraints, for all $s' \neq s_0, s_f$ and for all $x \in \mathcal{X}$, respectively:

$$\sum_{(s,s')\in\mathcal{A}} F(s \to s') = \sum_{(s',s'')\in\mathcal{A}} F(s' \to s''), \tag{4}$$

$$F(x \to s_f) = R(x). \tag{5}$$

We extend the definition of the flow function to states:

$$\forall s \in \mathcal{S}, \quad F(s) = \sum_{(s,s')\in\mathcal{A}} F(s \to s'). \tag{6}$$

We also define the forward and backward policies as follow:

$$P_F(s' \mid s) = \frac{F(s \to s')}{F(s)}, \quad P_B(s \mid s') = \frac{F(s \to s')}{F(s')}. \tag{7}$$

**Trajectory Balance (TB).** Given a complete trajectory/episode $\tau = (s_0 \to s_1 \to \ldots \to s_n = x)$ that terminates in $x$, the trajectory balance objective enforces a global balance using a learned normalizing constant $Z > 0$ (Malkin et al., 2022):

$$\mathcal{L}_{\text{TB}}(\tau) = \left( \log \frac{Z \prod_{t=0}^{n-1} P_F(s_{t+1} \mid s_t)}{R(x) \prod_{t=0}^{n-1} P_B(s_t \mid s_{t+1})} \right)^2. \tag{8}$$

Minimizing this loss across sampled trajectories ensures that $Z \prod_t P_F(\cdot) \approx R(x) \prod_t P_B(\cdot)$, which leads to sampling proportional to $R(x)$ under appropriate parameterizations.

**Sub-trajectory TB (SubTB).** To enable learning from partial episodes, SubTB (Madan et al., 2023) applies a TB-like constraint to partial trajectories $\tau_{0:k} = (s_0, \ldots, s_k)$:

$$\mathcal{L}_{\text{SubTB}}(\tau_{0:k}) = \left( \log \frac{F(s_0) \prod_{t=0}^{k-1} P_F(s_{t+1} \mid s_t)}{F(s_k) \prod_{t=0}^{k-1} P_B(s_t \mid s_{t+1})} \right)^2, \tag{9}$$

where $F(s)$ is the learned flow at state $s$.

In this work, we conducted our experiments using TB and SubTB losses for training GFlowNets. SubTB is particularly well-suited for our curriculum learning approach, as it naturally accommodates training on sequences of varying lengths and improves convergence for long mRNA sequences compared to standard TB loss.

## E    GFLOWNET TRAINING LOOP ALGORITHM

This section provides the detailed pseudocode for the GFlowNet Training Loop and the Curriculum-Augmented GFlowNet (CAGFN) training procedure, as described in Section 4.2. Algorithm 1 describes the former, while Algorithm 2 outlines the teacher-student interaction, task sampling, and learning progress updates.

### E.1 GFlowNet Training Loop

---

**Algorithm 1** GFlowNet Training Loop

---

**Require:** $\mathcal{E}$ (env), $\pi_\theta$ (policy), $R(\cdot)$ (reward), $B$ (batch), $I$ (iters), $\mathcal{O}$ (opt), $\boldsymbol{\alpha}$
1: **for** $i \in \{1, \ldots, I_{\text{total}}\}$ **do**
2: $\quad$ $\boldsymbol{w} \leftarrow$ NONE
3: $\quad$ **if** conditional **then**
4: $\quad\quad$ $\boldsymbol{w} \sim \text{Dirichlet}(\boldsymbol{\alpha})$
5: $\quad$ $\mathcal{T}_{1:B} \sim \pi_\theta(\cdot \mid \mathcal{E}, \boldsymbol{w})$ $\hfill \triangleright$ Sample $B$ paths
6: $\quad$ $r_{1:B} \leftarrow R(\mathcal{T}_{1:B})$
7: $\quad$ $\mathcal{L} \leftarrow \text{Loss}(\mathcal{E}, \mathcal{T}_{1:B}, r_{1:B})$ $\hfill \triangleright$ Eq. 9
8: $\quad$ $\theta \leftarrow \theta - \mathcal{O}(\nabla_\theta \mathcal{L})$
9: **return** $\pi_\theta$

---

## F Hyperparameters and implementation details

### Reward Details

In our implementation, we made use of previously published repositories, making small adaptations where necessary. We used the torchgfn library (Lahlou et al., 2023) to implement the GFlowNet model. To compute reward, we used 3 objectives : the **Codon Adaptation Index (CAI)**, a measure of how well the sequence conforms to species-specific codon usage preferences. Higher CAI typically leads to more efficient translation (Sharp & Li, 1987). To compute the CAI of mRNA sequences, we used the CAI library (Lee, 2018). The second objective is the **Minimum Free Energy (MFE)**, a (negative) folding free energy of the mRNA's secondary structure. A lower MFE (more negative) indicates a more stable folded structure. Then, third one is the **GC content,** which is the fraction of G or C nucleotides in the sequence. High GC-content generally correlates with increased mRNA stability and translation efficiency (Courel et al., 2019), in particular, optimizing GC-content has a similar effect to optimizing codon usage (Zhang et al., 2023).

### Dataset

We base our experiments on the dataset provided by the CodonTransformer paper (Fallahpour et al., 2025), which contains natural mRNA sequences corresponding to a diverse set of proteins. Each entry includes a DNA sequence, the corresponding protein sequence, and gene and organism information. Since our task focuses on mRNA design, we converted the DNA sequences into their corresponding mRNA sequences before using them in our experiments.

### Hyperparameters

This appendix lists the hyperparameters and implementation choices used for curriculum learning and the policy function (PF) in our GFlowNet experiments. Values shown are the defaults used in all experiments reported in the paper. We also briefly summarise the alternative ranges explored during tuning and the selection rationale.

Table 3: Curriculum learning hyperparameters

| Hyperparameter | Value |
|---|---|
| curriculum_tasks | [25,40], [45,60], [65,80], [85,120], [125,180] |
| n_iterations | 100 |
| eval_every | 5 |
| train_steps_per_task | 200 |

**Policy function (PF).** We choose for the PF module in the GFlowNets a Transformer architecture with the following hyperparameters: embedding_dim $= 32$, hidden_dim $= 256$, n_hidden $= 4$ Transformer layers, n_head $= 8$.

Table 4: Optimization and sampling hyperparameters

| Hyperparameter | Value |
|---|---|
| `lr_logz` | 1e-1 |
| `lr` | 5e-3 |
| `lr_patience` | 10 |
| `n_samples` | 100 |
| `top_n` | 50 |
| `batch_size` | 64 |
| `epsilon` | 0.25 |

Table 5: Teacher / curriculum selection hyperparameters

| Hyperparameter | Value |
|---|---|
| `lpe` (learning-progress estimator) | Online |
| `acp` (progress metric) | LP |
| `a2d` (action-to-difficulty) | GreedyProp |
| `a2d_eps` | 0.15 |
| `lpe_alpha` | 0.05 |
| `acp_MR_K` | 25 |
| `acp_MR_power` | 2 |
| `acp_MR_pot_prop` | 0.4 |
| `acp_MR_att_pred` | 0.1 |
| `acp_MR_att_succ` | 0.05 |

**Values explored during tuning.** During preliminary tuning, we explored multiple ranges for each major hyperparameter (final values are reported in the tables). For model capacity we tested `embedding_dim` $\in \{16, 32, 64\}$, `hidden_dim` $\in \{128, 256, 512\}$, and `n_hidden` $\in \{2, 4, 6\}$. For optimization we swept learning rates with `lr` $\in \{5 \times 10^{-4}, 1 \times 10^{-3}, 5 \times 10^{-3}\}$ and `lr_logz` $\in \{1 \times 10^{-2}, 1 \times 10^{-1}\}$. For the CL algorithm, we varied teacher parameters such as `lpe_alpha` $\in \{0.01, 0.05, 0.1\}$ and `acp_MR_K` $\in \{10, 25, 50\}$.

For each hyperparameter class we ran short preliminary experiments (a small number of curriculum iterations) to evaluate stability, wall-clock cost, and average learning progress. The reported defaults were selected as a trade-off between stable learning dynamics, empirical performance in these exploratory runs, and computational efficiency, when multiple configurations performed similarly we prioritized smaller/cheaper configurations. All experiments in the main text use the values listed in Tables 3–5.

## F.1 Curriculum Learning Configurations

To systematically evaluate the impact of different curriculum learning (CL) strategies on the multi-objective mRNA design task, we designed three distinct experimental configurations. These configurations are tailored to probe the trade-offs between learning stability, adaptiveness, and the exploration-exploitation balance. Each configuration uses a unique combination of components for Learning Progress Estimation (LPE), Attention Computation (ACP), and Attention-to-Distribution mapping (A2D), allowing us to isolate and analyze their effects on model performance.

**Configuration 1: Conservative EMA-based Curriculum.** This configuration is designed to prioritize stable, gradual learning, making it potentially more robust to the high variance often present in biological reward landscapes.

- Learning Progress Estimation (LPE): It employs an Online Exponential Moving Average (EMA) (lpe='Online') with a very small smoothing factor ($\alpha$=0.05). This ensures that the progress estimate is robust to noisy rewards and changes slowly, preventing drastic shifts in the curriculum.

- Attention Computation (ACP): Task selection relies on a simpler Learning Progress (LP) attention mechanism (acp='LP'), which directly uses the smoothed learning rate to prioritize tasks.

- Distribution Mapping (A2D): It uses an $\epsilon$-greedy proportional strategy (a2d='GreedyProp') with a moderate exploration rate of $\epsilon$=0.15. This forces the model to dedicate 15% of its time to uniform exploration, ensuring it does not prematurely converge on a suboptimal subset of tasks.

**Configuration 2: Aggressive Sampling-based Curriculum.** In contrast, this setup is engineered for rapid adaptation and aggressive exploration. It is intended to quickly discover and exploit promising areas of the vast mRNA design space.

- Learning Progress Estimation (LPE): It utilizes a Sampling-based progress estimator (lpe='Sampling') over a small, recent window of performance (K=10). This makes the curriculum highly responsive to immediate learning signals.

- Attention Computation (ACP): It leverages a more sophisticated Mastering Rate (MR) attention mechanism (acp='MR') with a high power parameter (power=8) and a strong emphasis on task potential (pot_prop=0.8). This aggressively prioritizes tasks that show the most promise for rapid improvement.

**Configuration 3: Balanced Proportional Curriculum.** This configuration seeks a robust balance between stability and adaptability, making it hypothetically well-suited for the complex trade-offs inherent in multi-objective optimization.

- Learning Progress Estimation (LPE): Learning progress is estimated using Linear Regression (lpe='Linreg') over a larger window (K=25). This provides a stable yet responsive measure of the learning trend, which is effective for handling noisy, multi-objective rewards by capturing the underlying slope of improvement.

- Attention Computation (ACP): It also uses the Mastering Rate (MR) attention mechanism but with moderate parameters (power=4, pot_prop=0.6) to balance focus between currently improving tasks and potentially valuable new ones.

- Distribution Mapping (A2D): Task distribution is purely proportional (a2d='Prop') to the computed attention weights. This represents a pure exploitation strategy based on the curriculum's policy, with no additional random exploration ($\epsilon$=0.0).

# G  CURRICULUM LEARNING ALGORITHM

---

**Algorithm 2** Curriculum Learning Algorithm

---

**Require:**

$\quad T = \{t_1, \ldots, t_N\}$ $\hfill \triangleright$ Pool of tasks sets where $t_i$ is a set tasks

$\quad S = \{S_1, \ldots, S_N\}$ $\hfill \triangleright$ Pool of protein sequence sets for each $t_i$

$\quad \pi_\theta$ $\hfill \triangleright$ GFlowNets Policy (Student)

$\quad \boldsymbol{w}_{\text{eval}}$ $\hfill \triangleright$ Fixed weights for eval

$\quad I_{\text{total}}, I_{\text{task}}, I_{\text{eval}}, \beta, \varepsilon$ $\hfill \triangleright$ Hyperparameters

1: $H_k \leftarrow \emptyset$ for each task $t_k \in T$ $\hfill \triangleright$ Initialize history for Learning Progress

2: $\mathcal{LP}_k \leftarrow 0$ for each task $t_k \in T$ $\hfill \triangleright$ Initialize Learning Progress for $t_i$

3: $P \leftarrow \text{Uniform}(1/N)$

4: **for** $i \in \{1, \ldots, I_{\text{total}}\}$ **do**

5: $\quad k \sim P, p_{\text{seq}} \sim S_k$ $\hfill \triangleright$ Sample task index $k$ and a protein

6: $\quad \mathcal{E} \leftarrow \text{InitializeEnv}(p_{\text{seq}})$

7: $\quad$ **for** $s \in \{1, \ldots, I_{\text{task}}\}$ **do**

8: $\quad\quad \boldsymbol{w} \sim \text{Dirichlet}(\boldsymbol{\alpha})$

9: $\quad\quad \mathcal{T} \sim \pi_\theta(\cdot | \mathcal{E}, \boldsymbol{w})$ $\hfill \triangleright$ Sample trajectory

10: $\quad\quad \mathcal{L}_{\text{SubTB}} \leftarrow \text{CalculateLoss}(\mathcal{T})$

11: $\quad\quad \theta \leftarrow \text{OptimizerStep}(\theta, \nabla_\theta \mathcal{L}_{\text{SubTB}})$

12: $\quad$ **if** $i \in \{I_{\text{eval}}, 2I_{\text{eval}}, \ldots\}$ **then**

13: $\quad\quad$ **for** each task $t_j \in T$ **do**

14: $\quad\quad\quad p_{\text{seq}} \sim S_j$

15: $\quad\quad\quad \mathcal{E}_{\text{eval}} \leftarrow \text{InitializeEnv}(p_{\text{seq}})$

16: $\quad\quad\quad \mathcal{T}_{eval} \sim \pi_\theta(\cdot | \mathcal{E}_{\text{eval}}, \boldsymbol{w}_{\text{eval}})$ $\hfill \triangleright$ Generate a batch of trajectories

17: $\quad\quad\quad r_j^{\text{new}} \leftarrow \sum_{\mathcal{T} \in \mathcal{T}_{eval}} R(\mathcal{T})/|\mathcal{T}_{eval}|$ $\hfill \triangleright$ Compute normalized reward

18: $\quad\quad\quad r_j^{\text{prev}} \leftarrow \begin{cases} H_{j=|H_j|}, & |H_j| > 0 \\ 0 & |H_j| = 0 \end{cases}$ $\hfill \triangleright$ Assign reward or set to 0 if history is empty

19: $\quad\quad\quad \Delta m_j \leftarrow r_j^{\text{new}} - r_j^{\text{prev}}$

20: $\quad\quad\quad \mathcal{LP}_j \leftarrow (1 - \beta)\,\mathcal{LP}_j + \beta\,\Delta m_j$ $\hfill \triangleright$ Update moving average of the LP $t_i$

21: $\quad\quad\quad H_j \leftarrow H_j \cup \{r_j^{\text{new}}\}$ $\hfill \triangleright$ Update the history

22: $\quad\quad$ **for** $j \in \{1, \ldots, N\}$ **do**

23: $\quad\quad\quad P(j) \leftarrow \max(0, \mathcal{LP}_j) + \varepsilon$

24: $\quad\quad P \leftarrow P / \sum_{k=1}^{N} P(k)$

25: **return** $\pi_\theta$

---

# H  ADDITIONAL PLOTS

## H.1  LOSS CURVES

We found that the Sub-Trajectory Balance (SubTB) loss function is critical for maintaining numerical stability for sequences longer than approximately 55 amino acids, as the standard Trajectory Balance (TB) loss consistently failed. All subsequent models are built upon these two findings.

We provide additional loss curves in Figures 9a–9c that compare the standard Trajectory Balance (TB) loss with the Sub-Trajectory Balance (SubTB) loss on sequences of length up to 75 amino acids (AA). The results clearly illustrate that TB fails to converge, with the loss plateauing at high values and exhibiting unstable behavior. In contrast, SubTB consistently converges to a stable solution, demonstrating its robustness for longer sequences.

Furthermore, we investigated whether the convergence issues with TB could be alleviated by changing model architectures or tuning hyperparameters. Specifically, we trained the model using MLP, LSTM, and Transformer architectures across a wide range of learning rates, batch sizes, and hidden dimensions. Across all settings, the TB loss exhibited the same instability, confirming that the problem is inherent to the TB formulation rather than to the architectural choice. These findings motivated our decision to adopt SubTB for all subsequent experiments.

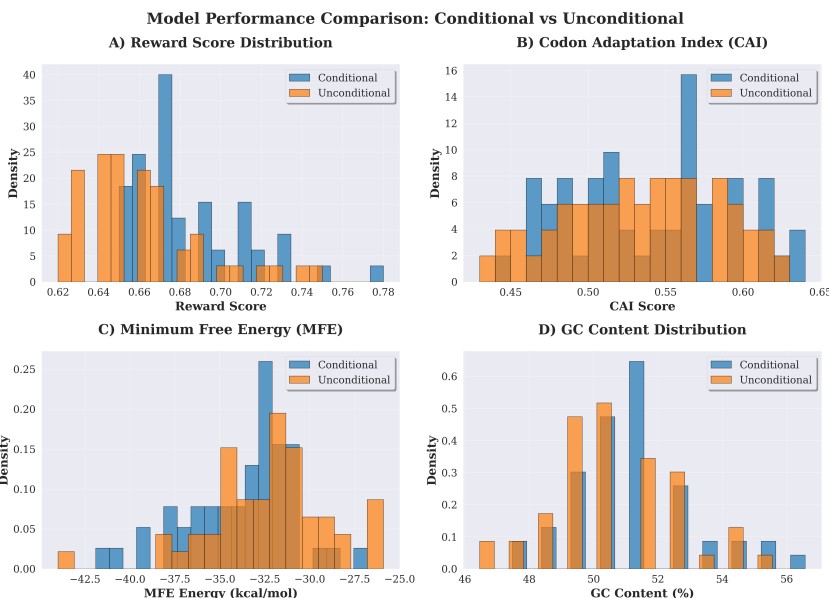

Figure 6: Metrics Distributions

In figure 10 : Panel (a) shows that the three non-curriculum baselines (*Short-Only*, *Long-Only*, *Random-Order*) fail to consistently reduce the loss and do not converge across training steps.

# I   EXTENDED RELATED WORK

GFlowNets are probabilistic generative models for sampling complex objects via sequential decisions. Unlike conventional RL that seeks a single optimal solution, GFlowNets train a policy to sample complete objects (e.g., sequences or graphs) with a probability proportional to a reward function. This proportional sampling intrinsically promotes diversity and allows GFlowNets to amortize the cost of sampling in large combinatorial spaces while representing distributions over composite structures (Bengio et al., 2021; 2023; Jain et al., 2022a; 2023). Empirically, GFlowNets have been applied to a range of scientific design tasks. For example, Jain et al. (2022a) embed a GFlowNet in an active learning loop to generate diverse candidate peptides and find more novel high-scoring sequences than previous methods. In drug discovery, several works extend GFlowNets to enforce domain constraints: Reaction-GFN (Koziarski et al., 2024) constrains generation to chemical reaction steps so that every molecule has a valid synthetic route, and SynFlowNet (Cretu et al., 2024) restricts actions to documented reactions and purchasable starting materials to produce synthetically accessible molecules while preserving diversity. These applications demonstrate that GFlowNets can flexibly incorporate domain-specific action spaces to generate complex, structured objects. Many design problems involve multiple, often conflicting objectives. Jain et al. (2023) have extended GFlowNets to handle multi-objective settings: they propose Multi-Objective GFlowNets (MOGFNs) to directly sample diverse Pareto-optimal solutions, and show in diverse experiments that MOGFNs achieve improved Pareto-front coverage and candidate diversity compared to standard RL baselines. Similarly, Zhu et al. (2023) integrate GFlowNets into a multi-objective Bayesian optimization framework via a hypernetwork-based GFlowNet (HN-GFN) that generates batches of molecules across different trade-offs, effectively sampling an approximate Pareto front. These works highlight that GFlowNets can be conditioned on objective preference vectors to explore trade-off spaces instead of collapsing to a single greedy objective.

GFlowNets have shown promise in molecular and peptide design, but their application to mRNA sequence design has not yet been explored and introduces distinct domain constraints. Although mRNA is a biological sequence, its design introduces unique objectives compared to typical molecular-generation tasks: codon choice (synonymous encodings), secondary-structure, and expression-related metrics must all be balanced. These domain-specific requirements make mRNA

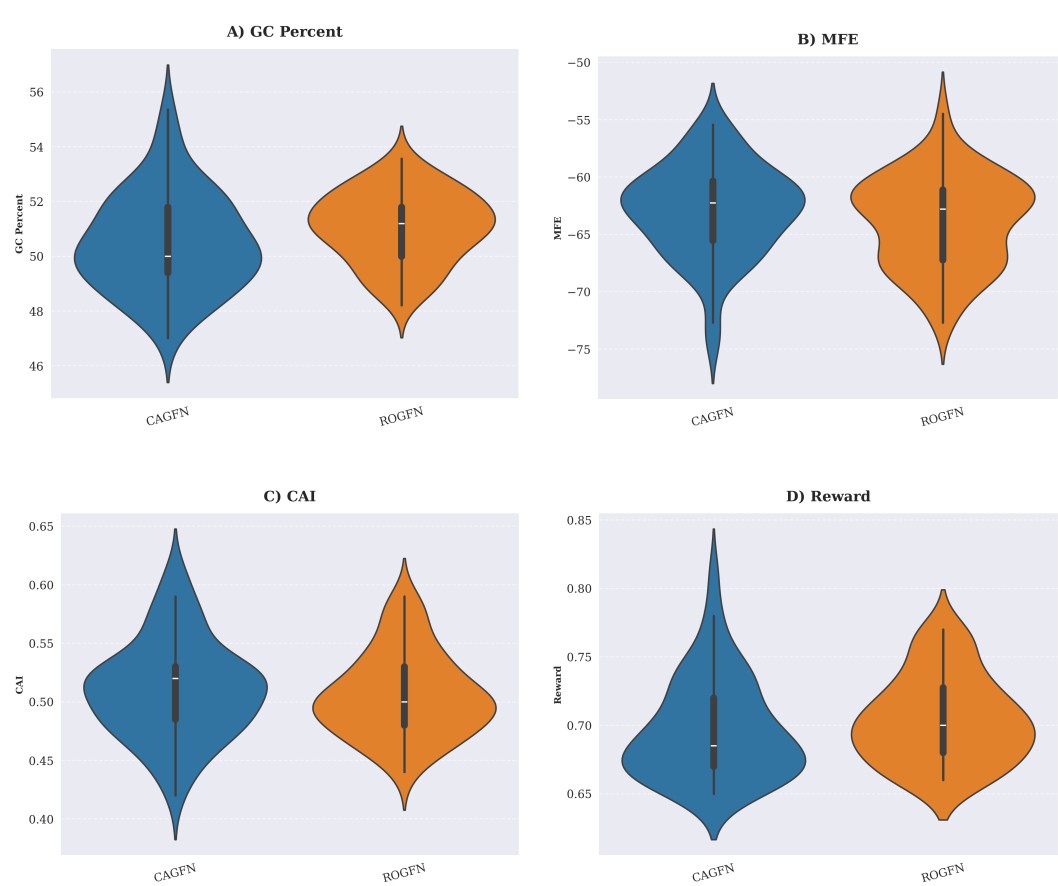

Figure 7: Distribution of reward and objectives for the medium-protein task, contrasting the curriculum-augmented GFlowNet (CAGFN) with a baseline GFlowNet trained without curriculum (ROGFN). Panels show empirical distributions of (a) Reward, (b) CAI, (c) MFE; lower values indicate more stable folding, and (d) GC content. Across all panels, CAGFN-shifted distributions indicate improved sampling: higher rewards and CAI, lower MFE, and GC content closer to the desired range [0.65, 0.35], suggesting that curriculum learning yields higher-quality and more biologically favourable mRNA candidates.

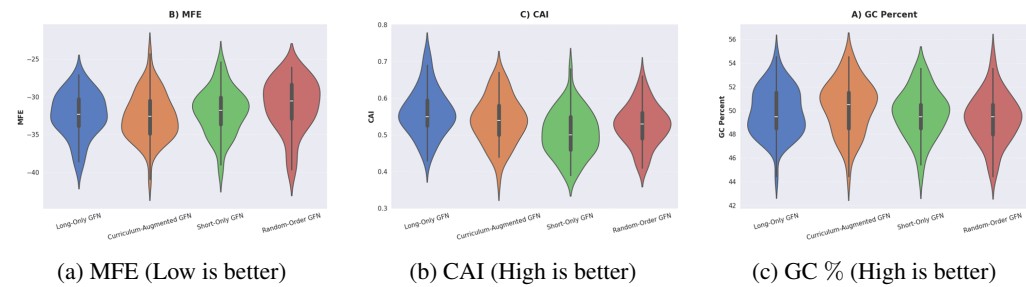

(a) MFE (Low is better)  (b) CAI (High is better)  (c) GC % (High is better)

Figure 8: Metrics distribution for the OAZ2 protein

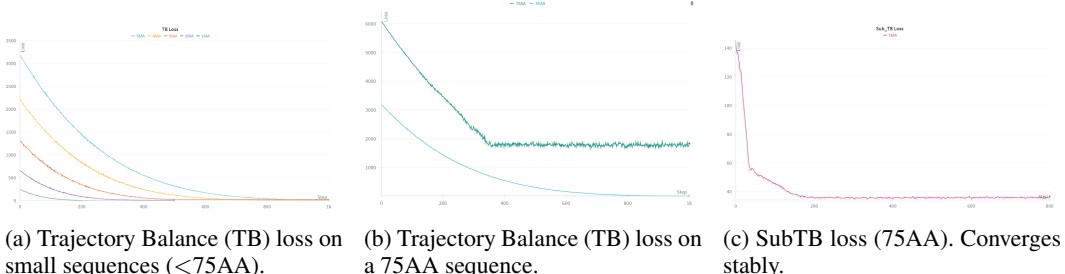

(a) Trajectory Balance (TB) loss on small sequences (<75AA).

(b) Trajectory Balance (TB) loss on a 75AA sequence.

(c) SubTB loss (75AA). Converges stably.

Figure 9: Sub-Trajectory Balance (SubTB) loss on 75AA sequences. Unlike TB, SubTB converges smoothly and remains numerically stable.

design a fundamentally different combinatorial problem from classical small-molecule generation, and they motivate adapting multi-objective GFlowNet methods to this setting.

The considerable length and structural complexity of mRNA sequences pose additional challenges for training GFlowNets, particularly with respect to credit assignment and sparse rewards. Curriculum learning addresses this by presenting tasks of increasing difficulty so that the model can master simpler subtasks first. Bengio et al. (2009) originally advocated for curriculum learning by training neural models on easier examples before harder ones. A prominent formalism in RL is Teacher–Student Curriculum Learning (TSCL; Matiisen et al., 2019), where a teacher dynamically selects tasks on which the student shows the fastest learning progress. In TSCL, the teacher monitors per-task learning curves and focuses on tasks with the steepest improvement (highest slope), which lets training rapidly move up in difficulty as the agent improves and has solved environments that were intractable under uniform sampling. The hypothesis we validate in this work is that applying an adaptive, learning-progress-driven curriculum to GFlowNets stabilizes credit assignment and accelerates convergence for long, sparse-reward sequence-design problems.

Taken together, these strands motivate our approach. We adopt the MOGFN formulation as the base model for our curriculum-augmented GFlowNet (CAGFN) to (i) generate diverse, Pareto-aware mRNA candidates and (ii) exploit a teacher-driven curriculum that progressively adapts sequence length during training based on measured learning progress. This combination addresses both the multi-objective nature of mRNA design and the training challenges posed by the length of these sequences.

## J  MRNA CODON DESIGN ENVIRONMENT

To exploit codon redundancy in protein coding, we introduce the **CODONDESIGNENV**, a discrete environment for codon-level mRNA design given a protein sequence. The environment models mRNA construction as a sequential decision process: states represent partially constructed sequences, actions correspond to choosing codons, and trajectories terminate once a full sequence encoding the target protein is completed.

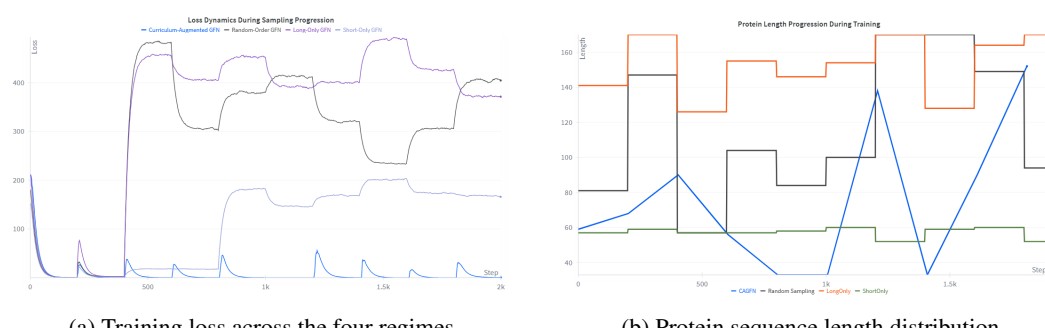

| (a) Training loss across the four regimes. | (b) Protein sequence length distribution. |

Figure 10: Loss dynamics and curriculum progression. (a) Training loss for *Short-Only*, *Long-Only*, *Random-Order*, and *Curriculum-Augmented GFN (ours)*. (b) Corresponding progression of protein sequence lengths shown to the model during training.

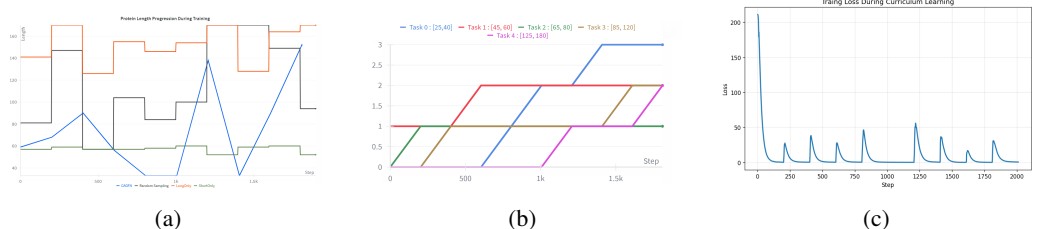

| (a) | (b) | (c) |

Figure 11: **Curriculum learning dynamics.** (a) We report the protein length progression of the four GFN variants. The comparison shows how the curriculum adapts the sequence length during training compared to alternative strategies. This curriculum schedule reduces the difficulty gap between tasks and allows the model to build competence over time without compromising simple tasks. (b) Indicates how often tasks of different lengths are sampled during curriculum, it shows the balance between exploration across easy and hard tasks, ensuring that the training process remains stable while covering the full range of sequence lengths. (c) CAGFN Loss. Brief spikes or oscillations may occur around curriculum tasks transitions.

**State Space ($\mathcal{S}$).** Each state $s \in \mathcal{S}$ is represented as a fixed-length integer vector of codon indices of size $L$ (protein length). Unassigned positions are denoted by $-1$, and the initial state is

$$s_0 = (-1, -1, \ldots, -1).$$

A state becomes terminal once all $L$ positions are filled with valid codons, yielding a complete mRNA sequence $x$.

**Action Space.** The action set consists of all $64$ codons plus a special "exit" action $a_{\text{exit}}$, i.e. $|\mathcal{A}| = 65$. At step $t$, the model selects a codon for the $t$-th amino acid in the target protein. Upon completion ($t = L$), the only valid action is $a_{\text{exit}}$, which transitions the environment to a terminal sink state $s_f$.

**Dynamic Masking Strategy.** To ensure biological validity, the environment uses dynamic action masking:

- **Synonymous codon masking:** At step $t < L$, the forward action set $\mathcal{A}_f(s)$ is restricted to codons encoding the $(t + 1)$-th amino acid.

- **Termination masking:** At step $t = L$, only $a_{\text{exit}}$ is permitted, forcing proper termination.

- **Backward masking:** For backward sampling, only the most recently added codon can be removed, preserving sequential construction.

These constraints define a directed acyclic graph (DAG) where each path from $s_0$ to $s_f$ corresponds to a valid mRNA sequence.

**Reward Function.** Rewards are assigned only to terminal states and capture biologically motivated objectives (§3.1): GC content, minimum free energy (MFE), and codon adaptation index (CAI). For a complete mRNA sequence $x$, the reward is

$$R(x) = w^\top \phi(x),$$

where $\phi(x)$ is a vector of normalized objectives and $w \in \mathbb{R}^3$ are user-defined weights. By adjusting $w$, one can bias the GFlowNet toward different Pareto-optimal trade-offs, promoting diversity and mode coverage across the design space.

While we adopt the weighted-sum formulation in this work for simplicity and interpretability, the same framework is compatible with alternative scalarization schemes. More generally, our approach follows the idea of *preference-conditional GFlowNets* (MOGFN-PC; Jain et al., 2023), in which a single reward-conditional GFlowNet models the entire family of multi-objective optimization subproblems simultaneously. Here, preferences $\omega \in \Delta^d$ over the set of objectives $\{R_1(x), \ldots, R_d(x)\}$ act as conditioning variables, and the reward function is defined via a scalarization $R(x \mid \omega)$. This general formulation accommodates any scalarization function, whether classical weighted sums, max–min formulations, or novel functions designed for specific biological contexts. For example, alternatives such as the weighted-log-sum scalarization have been proposed to mitigate cases where one objective dominates the reward signal.

# K  LLM USAGE

We used LLMs to aid and polish writing throughout this manuscript. Specifically, we employed large language models for grammatical refinement, clarity improvements, and ensuring consistency in technical terminology across sections. All scientific content, experimental design, results, and interpretations represent original work by the authors, with LLMs serving solely as writing assistants for language enhancement.

