# OpenReview forum: "Curriculum-Augmented GFlowNets For mRNA Sequence Generation"
_ICLR.cc/2026/Conference — Submitted to ICLR 2026_

### Official Review · Reviewer_E5FW · 2025-10-29

**Soundness:** 2
**Presentation:** 2
**Contribution:** 3
**Rating:** 4
**Confidence:** 3

**Summary:**

The authors tackle the problem of mRNA sequence design, where the task is to explore a vast space of nucleotide combinations while optimizing for properties such as stability, translation efficiency and protein expression. They identified GFlowNets as a promising candidate for this task and proposed curriculum-augmented GFlowNets (CAGFN) that integrates a length-based curriculum as well as a new mRNA design environment that trains the model to generate plausible mRNA candidates given a target protein sequence and a combination of biological objectives.

**Strengths:**

1. This paper gives very detailed and well-versed background on what the authors believe to be good mRNA sequence design: to generate diverse sequences that (1) preserve the desired protein sequence, (2) satisfy biological constraints, and (3) expose explicit trade-offs between competing objectives.
2. The authors give a fairly reasonable justification for using GFlowNets as a baseline for the development of the proposed method, namely that GFlowNet learn policies that sample complete objects with probability proportional to a user-specified reward, which supports the discovery of diverse solutions and naturally avoids mode collapse.
3. The authors nicely explained the limitations of directly applying GFlowNets on mRNA sequence design and proposed a remedy using curriculum learning.

**Weaknesses:**

1. The authors introduced three critical criteria for good mRNA sequence design (Strength 1) but the results do not seem to address them point-by-point. If the authors believe they have already done that, I would appreciate an explanation and potentially a revision to make that more obvious.
2. The results do not seem particularly impressive, with very mild improvements on codon adaptation index, minimum free energy, and GC content compared to a limited set of baselines. More importantly, since one of the main contributions of the paper is justifying an adapted version of GFlowNets for this task, it is worth demonstrating superior performance over other methods, not just variants of GFlowNets.
3. The presentation of the results can be further improved. For example, I would recommend showing the three metrics individually rather than only showing the reward which could be somewhat indirect.

**Questions:**

See Weaknesses.

---

> ### Author Response · Authors · 2025-11-20
>
> We thank the reviewer for the positive assessment and for highlighting the strengths of our work. We have carefully considered your comments to improve our paper.
>
> > W1. The authors introduced three critical criteria for good mRNA sequence design (Strength 1) but the results do not seem to address them point-by-point. If the authors believe they have already done that, I would appreciate an explanation and potentially a revision to make that more obvious.
>
> We agree that it is important to explicitly connect our results to the three key criteria for mRNA sequence design. In the revised paper, we will clarify this by referring to Figure 7, which compares the curriculum-augmented GFlowNet (CAGFN) with a baseline GFlowNet trained without curriculum (ROGFN) on the medium-protein task, and add similar plots. The panels show empirical distributions of:
>
> (a) Reward: higher values for CAGFN indicate improved overall sequence quality.
>
> (b) CAI: higher codon adaptation index reflects better optimization of translational efficiency.
>
> (c) MFE: lower values indicate more stable mRNA folding.
>
> (d) GC content: shifted closer to the desired range [0.35, 0.65], showing better biological suitability.
>
> Across all metrics, in figure 7, the distributional shifts for CAGFN demonstrate that the benefits of the curriculum learning. We will also enhance the discussion in the revised version to make this point explicit.
>
> >W2. The results do not seem particularly impressive, with very mild improvements on codon adaptation index, minimum free energy, and GC content compared to a limited set of baselines. More importantly, since one of the main contributions of the paper is justifying an adapted version of GFlowNets for this task, it is worth demonstrating superior performance over other methods, not just variants of GFlowNets.
>
> We evaluated GFlowNets against non-GFlowNet baselines earlier in the paper (in Figure 5). Specifically, in the initial experiment, we compared multi-objective GFlowNets with standard reinforcement learning methods (REINFORCE, PPO). We observed that RL baselines suffered from mode collapse and low diversity, generating repetitive sequences (88% duplicates for PPO).
>
> The choice of GFlowNets is consistent with prior work: both the MOGFN paper (Jain et al., 2023) and the GFlowNets for biological sequence design paper (Jain et al., 2022) have shown that GFlowNets outperform classical RL and heuristic methods in structured biological design spaces.
>
> Moreover, we did not use MCMC-based sampling because it is inefficient in high-dimensional discrete sequence spaces due to slow mixing and poor exploration. As previously discussed, GFlowNets amortize exploration costs over learning and scale more effectively than MCMC, making them a practical choice for mRNA design.
>
> > W3. The presentation of the results can be further improved
>
> In the revised version, we will improve the presentation of the results by showing the three main metrics individually, in addition to the overall reward.
> Additionally, we will include results from the Hypergrid benchmark (plots are available here : https://imgur.com/a/4fFeB4T) to further illustrate the benefits of curriculum strategies in a controlled, synthetic setting.

---

> > ### Comment · Reviewer_E5FW · 2025-11-28
> > **Response to rebuttal**
> >
> > I would like to thank the authors for the time and effort spent in the rebuttal.
> >
> > Weakness 1
> >
> > I am partially satisfied with the authors' response on weakness 1. I can acknowledge it is great that the authors show improvements on the four metrics (reward, CAI, MFE, and GC content). Meanwhile I am not quite persuaded that the authors sufficiently addressed this concern.
> >
> > I can see that "satisfy biological constraints" can be tied to "reward" which implies "sequence quality" and "MFE" which implies "stability". I can also see that "trade-offs between competing objectives" by optimizing four metrics simultaneously. However, it is a bit unclear how well these metrics reflect these implied features. For example, MFE is not a strong indicator of stability, and the "reward" seems a little bit arbitrary too.
> >
> > Weakness 2
> >
> > I agree that the baseline choice is somewhat consistent with the papers "Multi-Objective GFlowNets" and "Biological Sequence Design with GFlowNets", but it also needs to be recognized that many other works have been done since then, and GFlowNet variants are not the unanimous optimal solution in the field. There are multiple works based on GAN, VAE, diffusion models, flow matching, among many others. In general it would be great if the authors can show relative advantage of the proposed GFlowNets variant. This doesn't necessarily mean you need to beat all other methods in all benchmarks, but quantitative results that show some advantage (for example, better tradeoff of multiple objectives, more stable training, etc) can be beneficial.

---

> > > ### Author Response · Authors · 2025-12-01
> > >
> > > Thank you for your comment.
> > >
> > > Weakness 1.
> > >
> > > In our work we combined a multiple complementary metrics into a unified reward that captures sequence quality (CAI, GC codon usage), and structural stability ( MFE). This multi-factor reward is directly aligned with the established practice in de novo sequence design (e.g., Gu et al., 2023; Zhang, He, et al., Nature 621, 2023), where no single metric is a definitive proxy, but the joint optimization over multiple informative signals produces biologically meaningful sequences. We also emphasize that our goal is not to claim MFE alone is a stability predictor, but rather that when combined with GC content and CAI, GFlowNets combined with CL can expose and navigate the trade-offs between these coupled sequence-level constraints.
> > >
> > > Weakness 2.
> > >
> > > Our choice to build on GFlowNets is motivated by recent work, particularly Zhang et al. (2023), “Unifying Generative Models with GFlowNets and Beyond,” that showed the connections between existing deep generative models and GFlowNets. We therefore view GFlowNets as a unifying probabilistic framework and we test the curriculum learning integration for the mRNA design task.

---

### Official Review · Reviewer_2r1k · 2025-10-30

**Soundness:** 1
**Presentation:** 3
**Contribution:** 2
**Rating:** 2
**Confidence:** 3

**Summary:**

This paper proposes Curriculum-Augmented GFlowNets (CAGFN) that combine curriculum learning with multi-objective GFlowNets for de novo mRNA sequence generation. Specifically, the paper considers the task of generating multi-objective optimized mRNA condon sequences given a target protein sequence. The paper introduces adaptive, length-based curricula that preferentially present tasks (by protein length) where the model is demonstrating the fastest learning progress.

In empirical experiments, the paper first showed that multi-objective GFN achieves better diversity and rewards than other RL baselines, such as PPO, Reinforce. Then, the paper compared its curriculum strategy against three other baselines, including short-only, long-only, and random order, and showed that CAGFN trains significantly faster while achieving competitive results on Pareto performance and reward metrics.

**Strengths:**

1. The introduction of mRNA design for a specific target protein and the motivation for leveraging GFN for combinatoric discrete mRNA codon design is well written and clearly explained.

2. The proposed integration of curriculum learning and multi-objective GFN is a practicable and well-motivated approach for addressing the long-horizon, sparse-reward problem inherent in GFlowNet training for long sequences.

3. In the empirical experiments, CAGFN achieves much faster training while maintains competitive performances across different benchmarks.

**Weaknesses:**

1. The paper's core motivation is directly contradicted by its own experimental results. The paper's primary justification for a curriculum is that training on long sequences is difficult due to "extremely sparse credit assignment". However, the "Long-Only GFN" (LGFN) baseline, which was trained exclusively on these supposedly difficult long sequences, achieves the best or most competitive performance on the 85-120 AA generalization tasks (Table 1). This finding strongly suggests that the sparse reward problem is not as debilitating as claimed and that a curriculum is not necessary to achieve high performance.

2. CAGFN provides no clear performance benefit over non-curriculum baselines. The paper's central claims of "achieving higher rewards, better objective trade-offs, faster convergence, and broader Pareto-front coverage" are not sufficiently supported by the data. In Figure(Table?) 5, the full CAGFN model shows no performance advantage over the standard MOGFN without curriculum. In the main ablation (Table 1 & 2), CAGFN is not better than other baselines in majority cases, especially in 85-122 AA task where LOGFN is the best in 4 out of 5 metrics. In addition, the performance differences between different ablation baselines are quite small in general, especially considering the larger variances of CAGFN. These results indicate that the only demonstrable advantage of CAGFN is training speed over other curriculum baselines, not superior generative quality or generalization.

3. In terms of novelty, I appreciate the authors have acknowledged that the novelty of the paper is combining curriculum learning with GFN on the task of mRNA design. I would highly recommend exploring more meta-learning strategies beyond the simple 3 ablation baselines studied in the paper to explore the potential of curriculum-based methods to genuinely improve GFlowNet performance, rather than just training speed.

**Questions:**

While the idea is interesting and the training speedup is notable, the core claims are not convincingly supported by the paper's own empirical results (Tables 1, 2, and 5). The strong performance of the LGFN baseline, in particular, undermines the paper's central motivation. The advantages of the proposed method, therefore, appear to be limited to computational efficiency, which is a mismatch with the paper's stronger claims of improved generative performance.

For improvement, I would highly recommend exploring more meta-learning strategies beyond the simple 3 ablation baselines studied in the paper to explore the potential of curriculum-based methods.

In addition, although the paper compared with 3 curriculum strategies on 4 different protein metrics, there is no discussion on why a specific curriculum is better or worse on different metrics. A further analysis and interpretation on the results would greatly strengthen the paper by providing insight into how the curriculum is guiding the optimization process, rather than just presenting the final numbers.

---

> ### Author Response · Authors · 2025-11-20
>
> We thank the reviewer for their feedback and suggestions.
>
> > W1. The paper's core motivation is directly contradicted by its own experimental results. The paper's primary justification for a curriculum is that training on long sequences is difficult due to "extremely sparse credit assignment". However, the "Long-Only GFN" (LGFN) baseline, which was trained exclusively on these supposedly difficult long sequences, achieves the best or most competitive performance on the 85-120 AA generalization tasks (Table 1). This finding strongly suggests that the sparse reward problem is not as debilitating as claimed and that a curriculum is not necessary to achieve high performance.
>
> We want to clarify that LGFN was included as a challenging performance baseline to test whether simply training on long sequences suffices. While LGFN achieves competitive final performance in some metrics, it requires substantially more training.
> This aligns with prior work showing that sparse and delayed rewards in long sequential tasks create challenging credit assignment problems (Malkin 2022; Shen 2023). Crucially, under the same compute budget used in our main experiments, LGFN does not match the performance of CAGFN.
>
> In contrast, CAGFN achieves comparable or better performance while converging faster. These improvements in convergence speed and sample efficiency are critical for practical mRNA design, enabling more efficient exploration of the large sequence space and directly leading to higher-quality sequences within the same computational budget.
>
> > W2. CAGFN provides no clear performance benefit over non-curriculum baselines. The paper's central claims of "achieving higher rewards, better objective trade-offs, faster convergence, and broader Pareto-front coverage" are not sufficiently supported by the data. In Figure(Table?) 5, the full CAGFN model shows no performance advantage over the standard MOGFN without curriculum. In the main ablation (Table 1 & 2)...
>
> While LGFN achieves strong performance on the 85–122 AA evaluation, it does so at a much higher training cost, and under equal compute constraints it does not reach these results. In contrast, CAGFN achieves comparable or better performance while maintaining significantly higher sample efficiency and a faster convergence. These properties translate directly into discovering higher-quality mRNA sequences under realistic compute budgets.
>
> To further demonstrate the benefits of curriculum learning, we added experiments on the Hypergrid benchmark (8×8×8x8 and 128x128 grids) under three regimes: Baseline (no curriculum), Linear curriculum, and Sigmoid/exponential curriculum. Plots of the number of modes discovered and L1 distance to the target distribution show that curriculum strategies significantly improve exploration and convergence compared to the baseline. On the 8×8×8x8 grid, it finds all 16 modes by step ~100 (compared to only ~7 for the baseline at the same step and ~14 even by step 300), and all 4 modes by step 600 on the harder task 128x128 grid (versus ≈0–1 for the baseline even by step 1000).
>
> Additional plots are available here: https://imgur.com/a/4fFeB4T.
>
> > W3/Q1: Meta learning strategies
>
> We thank the reviewer for this suggestion. Curriculum learning and meta-learning address related but distinct problems. Curriculum learning controls the order and distribution of training tasks to make learning easier (we used the teacher–student scheduling). Meta-learning aims to learn how to adapt quickly to new tasks (learn-to-learn) by changing the model or optimization across tasks.
> While meta-learning strategies are indeed complementary and an interesting direction for future work, they are beyond the scope of the current paper.
>
> > Q2: A further analysis and interpretation :
>
> At each step, the teacher (Curriculum algorithm) selects a task and trains the student on sequences from that interval. Periodically, the student is evaluated across all tasks, and the teacher updates its task distribution based on performance.
>
> During this process, the mRNA properties sensitive to simpler sequence properties (e.g., GC content, codon adaptation index) improve quickly because shorter-length tasks are learned first. Metrics requiring long-range dependencies or exploration (e.g., sequence diversity and Minimum Free Energy) benefit from the gradual shift toward longer sequences, allowing the model to progressively capture complex dependencies.
>
> By progressively focusing the model’s effort where it can make the most progress, the curriculum ensures faster convergence, stable learning, and better overall sequence quality, rather than just accelerating training on average.
>
> We will include additional results and interpretations in the revised version to better illustrate how the curriculum guides the optimization of different metrics throughout training.

---

### Official Review · Reviewer_63Kj · 2025-11-01

**Soundness:** 2
**Presentation:** 2
**Contribution:** 1
**Rating:** 2
**Confidence:** 4

**Summary:**

This paper introduces Curriculum-Augmented GFlowNets (CAGFN), a method that integrates curriculum learning with multi-objective conditional GFlowNets to generate diverse and high-rewarded mRNA sequences conditionally on a given set of weights for each objective. The approach tackles long-horizon challenges of this task by using a length-based curriculum that guides the model to learn from an adaptive task distribution. CAGFN is evaluated on mRNA design tasks, a newly introduced environment.

**Strengths:**

1. Improving GFlowNet training for long-horizon settings is an important research question.
2. A new environment in the science domain is a good contribution for both GFlowNet and the scientific discovery community.

**Weaknesses:**

1. While the clarity is fine in general, there are some concerns. Most importantly, their usage of "conditional" or "unconditional" generation seems ambiguous. If I understand correctly, the mRNA environment is inherently conditional, conditioned on a target protein $p_{\text{seq}}$ with length $L$. The goal is to train a _single_ GFlowNet policy that generates sequences proportionally to a reward function, for each target, i.e., $P_{F}^{\top}(x;p_{\text{seq}}) \propto R(x;p_{\text{seq}})$ where $P_{F}^{\top}$ is marginal of $P_F$ over $\mathcal{X}$. If we additionally conditioned on the reward weights, the goal will be $P_{F}^{\top}(x; p_{\text{seq}},w) \propto R(x;p_{\text{seq}}, w)$. I was confused about this point when I first read the paper. I believe it would be clearer to explicitly state the target protein condition in section 4.1. Please correct me if I'm wrong. There are also some minor issues (see the "Minor" section below).
2. The method initialises the task distribution (Eq. (3)) as uniform over lengths, and I think this cannot be considered as "curriculum" learning, since it does not encourage progressive learning from easy (short) to harder (longer) tasks. And this contradicts the motivation of this work.
3. Empirical results are weak. 1) Specifically, standard benchmarks for GFlowNets (e.g., GridWorld, bit sequence generation) are missing, while the proposed curriculum learning method can be applied straightforwardly. 2) There's only a marginal performance gain over MOGFN (Figure 5) or ROGFN (Table 1) when considering the reported error range (standard deviation).

(Minor)
4. Line 66: TB does not reduce the variance compared to FM or DB. It's the opposite.
5. Line 190: $a$ is already used as an action in section 3.2.
6. Line 209: "Fig" is missing for the figure 1 reference.
6. Figure 2(a) is hard to digest; I have no idea how I can recognise the Pareto front from it.

**Questions:**

1. How's the distribution of $L$ if you randomly sample the target protein from the environment?

---

### LLM usage disclosure
I did not use an LLM for my review, but I used grammar check software.

---

> ### Author Response · Authors · 2025-11-20
>
> Thank you for your feedback. We have carefully considered both your comments to improve the clarity and quality of the paper.
>
> > W1. While the clarity is fine in general, there are some concerns. Most importantly, their usage of "conditional" or "unconditional" generation seems ambiguous...
>
> The key distinction for the protein conditioning is:
>
> - The environment is indeed conditional on the target protein sequence: for each protein p, the reward function $R(x,p)$ and the valid action space depend on p.
>
> - The GFlowNet policy, however, is not explicitly conditioned on p (we do not feed a protein embedding to the policy network). Instead, the GFlowNet learns biological rules and constructs the flow function without seeing the full protein sequence. The implicit conditioning (what made the released ambiguity) arises through the interaction with the environment: at each training step, the GFlowNet receives trajectories and rewards for the currently sampled protein target.
>
> > W2. The method initialises the task distribution (Eq. (3)) as uniform over lengths...
>
> We only initialize the task distribution as uniform at the beginning. After that, it is updated dynamically according to the learning progress of the GFlowNet (figure 10b), following the teacher–student mechanism described in 4.2. Each “task” corresponds to a protein-length interval, and the teacher updates the probabilities based on the learning progress of the student, as described in the Algorithm 2, inspired by (Matiisen, Tambet, et al. 2017).
>
> > W3. Empirical results are weak...performance gain over MOGFN (Figure 5) or ROGFN (Table 1) when considering the reported error range (standard deviation).
>
> We provide additional clarification of our empirical results. Figure 5 presents a sanity check comparing CAGFN with MOGFN on a small protein task, verifying that:
>
> > - Performance parity: CAGFN matches a task-specific MOGFN.
>
> > - No forgetting: curriculum learning does not harm performance on simpler tasks.
>
> While MOGFN is trained exclusively on this small protein task and cannot generalize to other proteins, CAGFN is trained using CL on different protein lengths (generalist). Despite this broader training, CAGFN achieves similar performance on the small task while retaining the ability to generalize to new proteins.
>
> Also, GFlowNet variants achieve higher uniquenes and diversity, outperforming RL baselines and confirming prior observations.
>
> Our design isolates the effect of curriculum learning:
>
> - SGFN: fast training on short sequences but limited quality
> - LGFN: learning long sequences from scratch is expensive
> - ROGFN: diverse data alone is insufficient, task order matters
> - CAGFN: curriculum-ordered training provides the best trade-off between quality and efficiency
>
> Thus, the main benefits of curriculum learning are faster training and more efficient exploration, which translate into discovering high-quality, diverse sequences within the same computational budget. We will clarify these points in the paper.
>
> We also acknowledge that the presentation of these results could be clearer. In the revised version, we will improve the figures and tables to better highlight performance differences, generalization, and the benefits of curriculum learning.
>
> > W3. 1) Standard benchmarks for GFlowNets
>
> To demonstrate the benefits of curriculum learning, we added experiments on the Hypergrid benchmark (8×8×8x8 and 128x128 grids) under three regimes: Baseline (no curriculum), Linear curriculum, and Sigmoid/exponential curriculum. Plots of the number of modes discovered and L1 distance to the target distribution show that curriculum strategies significantly improve exploration and convergence compared to the baseline. On the 8×8×8x8 grid, it finds all 16 modes by step ~100 (compared to only ~7 for the baseline at the same step and ~14 even by step 300), and all 4 modes by step 600 on the harder task 128x128 grid (versus ≈0–1 for the baseline even by step 1000), (Plots: https://imgur.com/a/4fFeB4T).
>
> > Q1. How's the distribution of L if you randomly sample the target protein from the environment?
>
> The target protein is not randomly sampled from the environment. Instead, the curriculum defines a distribution over tasks (protein-length intervals) :  at each step, a task is sampled according to this distribution, and a protein length is drawn uniformly from the interval. Thus, the protein length distribution is entirely determined by the curriculum.
>
> We thank the reviewer for the detailed feedback on minor issues. We corrected the following points in the revised version:
>
> - Line 66: we corrected the TB behavior in the revised version.
> - Line 209: Thank you for noticing. We added it in the revised version.
> - Line 190 (reuse of a): We acknowledge the notation overlap. Here, 𝑎 represents both an action and its corresponding amino acid, which is why it appears twice.
> - Figure 2(a) readability: We will update the figure to better highlight the Pareto front.

---

> > ### Comment · Reviewer_63Kj · 2025-11-20
> >
> > Before adding comments on other points, let me quickly ask few questions regarding the following point:
> > > The environment is indeed conditional on the target protein sequence: for each protein p, the reward function $R(x,p)$
> >  and the valid action space depend on p. The GFlowNet policy, however, is not explicitly conditioned on p (we do not feed a protein embedding to the policy network)...
> >
> > 1. Why didn't you use the protein embedding?
> > 2. Under this setting, what is the optimal policy? To me, it seems impossible to optimize the loss to zero, so the optimally learned model is not able to sample proportionally to $R(x;p,w)$ for any given $p$ and $w$. Is this correct?
> > 3. In Figure 10b, for baselines, why does the protein length change as a step function, while CAGFN's is a piecewise linear function?

---

> > > ### Author Response · Authors · 2025-11-20
> > >
> > > > Q1. Why didn't you use the protein embedding?
> > >
> > > In our setup, the environment already encodes the target protein sequence in the action space: the amino acid sequence determines which codons are available, and invalid codons are masked out. So, the policy network already implicitly “sees” the protein through this masking structure, for this reason, we think that adding an explicit protein embedding would add a redundant information.
> > > In preliminary experiments, we did try conditioning the policy directly on a protein embedding. However, this led to very high dimensional conditioning vectors (especially for long proteins) that are kept fixed along the whole trajectory, which made optimization of the GFlowNet even more difficult.
> > >
> > > > Q2. Under this setting, what is the optimal policy? ...
> > >
> > > The loss function in the figure 10 a) and 11 c) shows how the curriculum learning helped the GFlownets to achieve lower loss during training.  CAGFN exhibits brief spikes in loss whenever the curriculum introduces a harder length range, followed by a rapid decrease back to a low loss. This pattern indicates that the model reuses what it learned on shorter sequences to adapt to slightly longer ones, keeping training stable and avoiding the persistent high‑loss regime observed in the non‑curriculum baselines in figure 10 a). The curriculum learning primarily makes the optimization easier and more stable. Thus the CAGFN finds the optimal policy for any given weight vector (because of the conditioning) and also for new proteins because it learned the biological rules for obtaining high reward mRNA sequences.
> > >
> > >
> > > > Q3. In Figure 10b, for baselines, why does the protein length change as a step function, while CAGFN's is a piecewise linear function?
> > >
> > > For SGFN, LGFN, and ROFN the training distribution over sequence lengths is fixed, and does not change during regimes (after a number of steps). So, the “training sequence length” is constant within each regime and only changes abruptly when: the data loader reshuffles, and a new protein is sampled from the same pool.
> > > But for the CAGFN, and with the curriculum over length: as training proceeds, the curriculum gradually increases the probability of sampling longer proteins instead of jumping directly from “short” to “long”.
> > > In the Algorithm 2, the EMA‑based update of P (lines 19–24), combined with sampling from P (lines 4–5), produces a smooth piecewise linear increase in sequence length over training, unlike the baselines whose length distributions are fixed and thus appear step‑like.

---

### Official Review · Reviewer_d3S2 · 2025-11-02

**Soundness:** 3
**Presentation:** 3
**Contribution:** 2
**Rating:** 6
**Confidence:** 3

**Summary:**

The paper proposes a new GFlowNet algorithm for generating mRNA sequences. Since mRNA sequences are relatively long, the search space is extremely large, making it computationally demanding to train an effective flow function. To address this challenge, the paper introduces a curriculum learning approach that splits the training of the flow function into multiple tasks. The training begins with easier tasks and gradually progresses to more difficult ones. This gradual learning process helps the model develop a better flow function, as it can leverage knowledge gained from simpler tasks to generate more complex parts of the sequence. Experiments on multi-objective mRNA generation is conducted to showcase the effectiveness of the proposed approach.

**Strengths:**

- Employing GFlowNets for biological sequence generation has shown well-documented performance. However, training a flow function for relatively long sequences remains a significant challenge. This paper proposes a method to address this important issue.
- The use of curriculum learning to achieve this goal is well-motivated.

**Weaknesses:**

Although the idea of using curriculum learning is well-motivated, I believe the current implementation in the paper may not be as effective as expected and may lead only to marginal improvements. Based on my understanding, the training procedure is largely the same as in other GFlowNet methods, with the main difference being that the order of learning each part of the sequence is determined by inferred difficulty. This modification alone is unlikely to significantly improve the efficiency of GFlowNet training. I believe this is reflected in the experimental results, as the performance gains appear to be quite marginal.

**Questions:**

If the proposed method indeed splits sequence generation into tasks, with each task corresponding to generating a specific part of the sequence, then since the model generates sequences autoregressively, it would be expected that the most difficult tasks are generating the later parts of the sequence. Is this in agreement with your experimental observations?

---

> ### Author Response · Authors · 2025-11-20
>
> We thank the reviewers for their valuable feedback. In response, we revised the paper to clarify key aspects of the method and provide additional evidence. Below we address W1 and Q1.
>
> > W1: Although the idea of using curriculum learning is well-motivated, I believe the current implementation in the paper may not be as effective as expected and may lead only to marginal improvements. Based on my understanding, the training procedure is largely the same as in other GFlowNet methods, with the main difference being that the order of learning each part of the sequence is determined by inferred difficulty. This modification alone is unlikely to significantly improve the efficiency of GFlowNet training. I believe this is reflected in the experimental results, as the performance gains appear to be quite marginal.
>
> The goal of CAGFN is to improve GFlowNet training for mRNA design, where the search space grows exponentially with protein length and long sequences are significantly harder to learn. Unlike standard GFlowNets, our method uses a teacher–student curriculum over protein-length ranges.
>
> To do this, the teacher holds:
> > - T task: the length ranges.
> > - D: probability distribution over tasks (how likely we are to train on each range next).
> > - LP: learning progress signal of the student (GFlowNets)
>
> > At each Global step:
> 1.	Sampling a task: the algorithm draws a task index according to D. In the beginning, D is uniform across all tasks, then updated depending on the learning progress of the students.
> 2.	Train the student (GFlowNets) on a protein sequence from the chosen task.
> 3.	Then the student is evaluated on all tasks (after the training) and the teacher store performance history per task, to choose the next task based on its performance.
> 4.	Update the tasks distribution D: The probabilities for tasks that are underperforming (too hard right now) are increased and for tasks that are already mastered, the probabilities are reduced. The progress of the student defines the order of the tasks.
>
> This produces a training schedule that differs from vanilla GFlowNets and leads to both faster learning and better exploration, which in turn improves the quality of the generated sequences.
>
> To further demonstrate the gains, we ran additional experiments on the Hypergrid benchmark across both the 8×8×8x8 and 128x128 grids. On the 8×8×8x8 grid, CAGFN finds all 16 modes by step ~100, whereas the baseline needs nearly three times more training and still reaches only ~14 modes by step 300, showing a clear improvement in exploration efficiency. (Plots: https://imgur.com/a/4fFeB4T).
>
> > Q1: If the proposed method indeed splits sequence generation into tasks, with each task corresponding to generating a specific part of the sequence, then since the model generates sequences autoregressively, it would be expected that the most difficult tasks are generating the later parts of the sequence. Is this in agreement with your experimental observations?
>
> Our curriculum does not split sequences into autoregressive segments. Instead, each “task” corresponds to a protein-length range. Consequently, harder tasks are not later positions in a sequence but longer protein sequences, which we empirically observe to have slower learning progress, precisely what the curriculum targets.

---

### Author Response · Authors · 2025-12-04
**Rebuttal Summary**

The goal of CAGFN is to improve GFlowNet training for mRNA sequence design, where the search space grows exponentially with protein length and long sequences are significantly harder to learn. Unlike standard GFlowNets, our method uses a teacher–student curriculum over protein-length ranges.

To do this, the teacher holds:

- T task: the length ranges.

- D: probability distribution over tasks (how likely we are to train on each range next).

- LP: learning progress signal of the student (GFlowNets)

At each Global step:

- Sampling a task: the algorithm draws a task index according to D. In the beginning, D is uniform across all tasks, then updated depending on the learning progress of the students.

- Train the student (GFlowNets) on a protein sequence from the chosen task. Then the student is evaluated on all tasks (after the training) and the teacher store performance history per task, to choose the next task based on its performance.

- Update the tasks distribution D: The probabilities for tasks that are underperforming (too hard right now) are increased and for tasks that are already mastered, the probabilities are reduced. The progress of the student defines the order of the tasks.

This produces a training schedule that differs from vanilla GFlowNets and leads to both faster learning and better exploration, which in turn improves the quality of the generated sequences.

To further demonstrate the gains, we ran additional experiments on the Hypergrid benchmark across both the 8×8×8x8 and 128x128 grids. On the 8×8×8x8 grid, CAGFN finds all 16 modes by step ~100, whereas the baseline needs nearly three times more training and still reaches only ~14 modes by step 300, showing a clear improvement in exploration efficiency. (Plots: https://imgur.com/a/4fFeB4T).

We also clarified some reviewer misunderstandings in our detailed comments and highlighted the key contribution and important changes.

---

### Meta-Review · Area_Chair_vYkc · 2026-01-06

**Summary:**

The paper proposes integrating curriculum learning with multi-objective GFlowNets for mRNA sequence design. The method uses a teacher-student framework that adaptively schedules training on protein sequences of increasing length, addressing the challenge of sparse rewards in long-horizon sequence generation.

**Reviewer Concerns:**

Addressed concerns:

- Clarification on splitting sequences into parts for curriculum.
- Additional Hypergrid experiments for determining curriculum benefits.
- Clarification on the distinction between environment conditioning and policy conditioning.

Outstanding concerns:
- Limited performance gains (Reviewers 63Kj, 2r1k, E5FW)
- Novelty concerns (Reviewer 63Kj)
- Baseline comparisons (Reviewer E5FW)

**Reviewer Scores:**

Reviewers 63Kj, 2r1K, E5FW would not have changed their scores to be positive. While the paper addresses a relevant problem and the mRNA design environment is a useful contribution, the core claims about improved generative performance are not convincingly supported. The methodological novelty is limited, and the strong performance of simpler baselines (particularly LGFN) weakens the motivation for the proposed approach.

---

### Decision · Program_Chairs · 2026-01-26

Reject